# Hierarchical crack-resistant, tissue-mimetic hydrogels enabled by progressive nanocrystallization of anisotropic polymer networks

Huamin Li, Haidi Wu, Cheng Guan, Wenjie Hu, Wenwen Su, Dingdong Chen & Jiefeng Gao ✉

Addressing the persistent challenge of reconciling extreme mechanical robustness with tissue-mimetic functionality in hydrogels, we present a phase-transition-guided hierarchical engineering strategy that progressively architectures anisotropic polyvinyl alcohol networks through sequential mechanical training, wet-annealing, and salting-out. This triphasic processing induces programmable structural evolution: (1) mechanical training aligns polymer chains, (2) wet-annealing relaxes the stress while stabilizes oriented crystallites through solvent-plasticized rearrangement, and (3) salting-out densifies the network via chain aggregation and hydrogen-bond proliferation. The resultant hierarchical architecture achieves high fatigue resistance (threshold: 2083 J·m$^{-2}$) through multi-scale energy dissipation: sacrificial hydrogen bonds consume energy, while aligned crystalline domains pin the crack and deflect crack propagation via anisotropic stress redistribution. Demonstrating tissue-surpassing mechanics (tensile strength: 61 ± 3 MPa, toughness: 106 ± 27 MJ·m$^{-3}$, fracture energy: 85 ± 9 kJ m$^{-2}$) coupled with biological functionality, the hydrogel directs cell alignment through contact guidance while resisting swelling-induced dimensional instability (<1.2% volume change in physiological saline). This biomimetic engineering strategy establishes a universal route to design synthetic extracellular matrices that concurrently emulate the anisotropic mechanics of tendons and crack-blunting resilience of cartilage, critical for load-bearing tissue regeneration.

Hydrogels are three-dimensional network materials composed of hydrophilic polymers that form through physical or chemical cross-linking[1,2]. Their distinctive 'soft-wet' nature enables them to absorb and retain substantial amounts of water while maintaining structural integrity[3,4]. This architecture imparts biomimetic properties, including tissue-like softness, permeability, and elasticity, with modulus in the kilopascal range—closely resembling those of human soft tissues such as skin and cartilage[5,6]. Additionally, some hydrogels exhibit excellent biocompatibility[7–9], mechanical adaptability[10–12], and functional programmability[13], making them promising candidates for applications in tissue engineering[14,15], soft robotics[16,17], and environmental engineering[18–20]. However, their often inherently nonuniform polymeric network results in poor mechanical properties, including insufficient strength, low toughness, and inadequate fatigue resistance.

Inspired by natural structural tissues such as cartilage and tendons, strong anisotropic hydrogels with macromolecular alignment

School of Chemistry and Materials, Yangzhou University, Yangzhou, Jiangsu, China. ✉e-mail: jfgao@yzu.edu.cn

have been developed[21,22]. Unidirectional freezing[23,24] and mechanical stretching[25] (or shearing) are the most commonly employed methods for preparing anisotropic hydrogels. During unidirectional freezing, macromolecules separate from the solvent and align within the interstices of unidirectionally grown ice crystals. However, the resulting anisotropic gels typically exhibit weak inter-molecular interactions and poor mechanical properties. To address this, unidirectional freezing is often combined with salting out or hot-pressing to densify the polymeric network and enhance mechanical strength[26,27]. Additionally, pre-stretching and mechanical training effectively align macromolecular chains, generating fibrous bundles with biomimetic tissue-like structures[28,29]. For instance, polymer gels are first pre-stretched, then dried and annealed, preserving the tensile stress-induced anisotropic structure upon reswelling[30]. In many cases, salting out or annealing is used to further lock macromolecular chain alignment[31]. In crystalline macromolecules, stretching also enhances crystallinity and promotes the alignment of crystalline regions[32]. Repeated stretching mimics mechanical training, a process that disrupts the original nanofibrillar structures while generating new nanofibrils, similar to skeletal muscle remodeling[33]. Likewise, mechanical training of hydrogels can induce new nanocrystalline domains and fibrous bundles by reorganizing randomly distributed nanocrystalline domains and macromolecular chains[14]. Notably, mechanical training is typically conducted in a water bath to prevent water evaporation from the hydrogels[34].

Despite significant progress in anisotropic hydrogels, achieving simultaneous enhancement of compressive mechanical properties remains a challenge. As load-bearing materials, hydrogels must exhibit not only high strength but also superior toughness and fatigue resistance. However, strength and toughness are often mutually exclusive properties[22]. For instance, increasing the stretch ratio generally enhances tensile strength and modulus but reduces fracture strain, leading to a potential decline in toughness[35,36]. Moreover, high tensile strength does not necessarily correlate with improved fatigue resistance, necessitating the incorporation of multiple energy dissipation mechanisms to enhance crack propagation resistance[37]. Furthermore, anisotropic hydrogels must demonstrate swelling resistance and biocompatibility to maintain structural integrity and functional stability in physiological aqueous environments[30,38].

We present a progressive reinforcement strategy to engineer anisotropic hydrogels with mechanical synergy: simultaneous strength ($61 \pm 3$ MPa), toughness ($106 \pm 27$ MJ m$^{-3}$), fracture energy ($85 \pm 9$ kJ m$^{-2}$), and fatigue resistance (-2083 J m$^{-2}$), via the integration of mechanical training, wet-annealing, and salting-out. The presence of the high-boiling-point solvent glycerol facilitates efficient stress transfer and chain mobility, significantly enhancing orientation compared to traditional hydrogel training in water, which is slower, more complex, and less effective at inducing molecular alignment. In addition, conventional dry-annealing of hydrogels (e.g., after water removal or aerogel formation) cannot sufficiently activate macromolecular mobility, which restricts the improvement of crystallinity and chain entanglement. In contrast, confined wet-annealing of organogels provides a more favorable environment for chain rearrangement and conformational adjustment, as the presence of solvent reduces resistance to chain motion. This process thus promotes stronger hydrogen bonding and higher crystallinity, while salting-out induces hierarchical aggregation of aligned chains, forming a dual-reinforced network with crystallinity-enhanced domains and hydrogen-bond-densified interfaces. The stepwise combination of mechanical training, wet annealing, and salting-out produces a progressive, cooperative reinforcement of the hydrogel network. Mechanical training aligns the chains, wet annealing optimizes crystallinity and chain entanglement in amorphous regions (compensating for deformation loss due to orientation), and salting-out further consolidates chain aggregation and network density, locking in the anisotropic structure. This indispensable multistep synergy, which has

not been reported previously, enables the hydrogel to achieve good mechanical performance together with swelling resistance. This hierarchical architecture achieves a 25-fold increase in tensile strength compared to isotropic counterparts, while maintaining anti-swelling stability (<1.2% volume change in physiological fluids). Crucially, the fatigue threshold surpasses most synthetic hydrogels, attributed to crack deflection at aligned nanofibrils and energy dissipation via sacrificial hydrogen bonds. Beyond mechanical robustness, these hydrogels exhibit excellent biocompatibility (or cytocompatibility) (NIH/3T3 viability > 98%), anti-swelling properties, and the ability to guide directional cell alignment, making them highly promising for tissue engineering and bioelectronic applications.

## Results
### Design of anisotropic hydrogels
The hierarchical fabrication of anisotropic PVA hydrogels employs a phase-transition-guided strategy that progressively engineers multi-scale architectures through progressive nanocrystallization engineering of anisotropic polymer networks (Fig. 1a). Initial pre-freezing establishes a metastable hydrogen-bonded network, while glycerol/ethanol solvent exchange induces macromolecular aggregation via poor solvent effects[39], nucleating primitive crystalline domains. Subsequent mechanical training (30 cycles at a strain of 300% and a tensile rate of 200 mm/min) under controlled ethanol volatilization triggers alignment of PVA chains along the stress axis, with residual glycerol acting as a plasticizer to facilitate molecular slippage and orientation. Wet annealing in glycerol at 90 °C or 120 °C for 30 min enables defect repair through solvent-assisted chain rearrangement, densifying the network via increased crystallinity and H-bond density. Critical stabilization arises from Hofmeister-specific ion effects interactions during salting-out with 1.5 M sodium citrate solution, where citrate anions screen electrostatic repulsions between aligned chains, promoting hydrophobic collapse that locks the anisotropic architecture. This multi-stage process, combining solvent-mediated phase transitions (freezing/exchange/evaporation) with strain-crystallization coupling, creates a hierarchically ordered network with energy-dissipative mechanisms across multiple length scales, from molecular-level hydrogen bond rupture to microstructural crystalline domain slippage and macro-scale stress redistribution within aligned fiber bundles. Final aqueous equilibration preserves ion-induced chain aggregation and alignment, yielding transparent hydrogels (Fig. S1) with biologically mechanical synergy. The transparency of AV$_{15}$-120 is slightly lower than that of AV$_{15}$, which can be attributed to the crystallinity increase induced by annealing, whereas AV$_{15}$S-120 exhibits a much larger decrease in transparency. This pronounced reduction is ascribed to salting-out–induced chain aggregation, which can be regarded as a form of partial phase separation (Fig. S2). These hydrogels exhibit tendon-like strength (61 MPa), capable of lifting 12,000 times their own weight without failure (Fig. 1b), cartilage-mimetic crack resistance (Γ = 2083 J·m$^{-2}$), and pronounced anisotropy (orientation factor > 0.8, as observed by WAXS) for directional biomimicry. Compared to pure PVA hydrogels prepared by solvent exchange and salting out, the tensile strength and modulus are enhanced nearly 25-fold and 60-fold, respectively (Fig. 1c). The mechanical properties surpass those of certain human joint ligaments[40] (Fig. 1d) and approach tendon-level performance, highlighting their potential for hydrogel biofunctionalization in biomedical applications.

### Mechanical properties
Figure 2 systematically shows the mechanical performance of PVA hydrogels engineered through sequential phase-transition control, revealing a processing-structure-property nexus governed by the experimental parameterization. For clarity in descriptions and discussions, the following designations are used: IV$_x$SH refers to isotropic hydrogels prepared via wet-annealing and salting out; AV$_x$ denotes

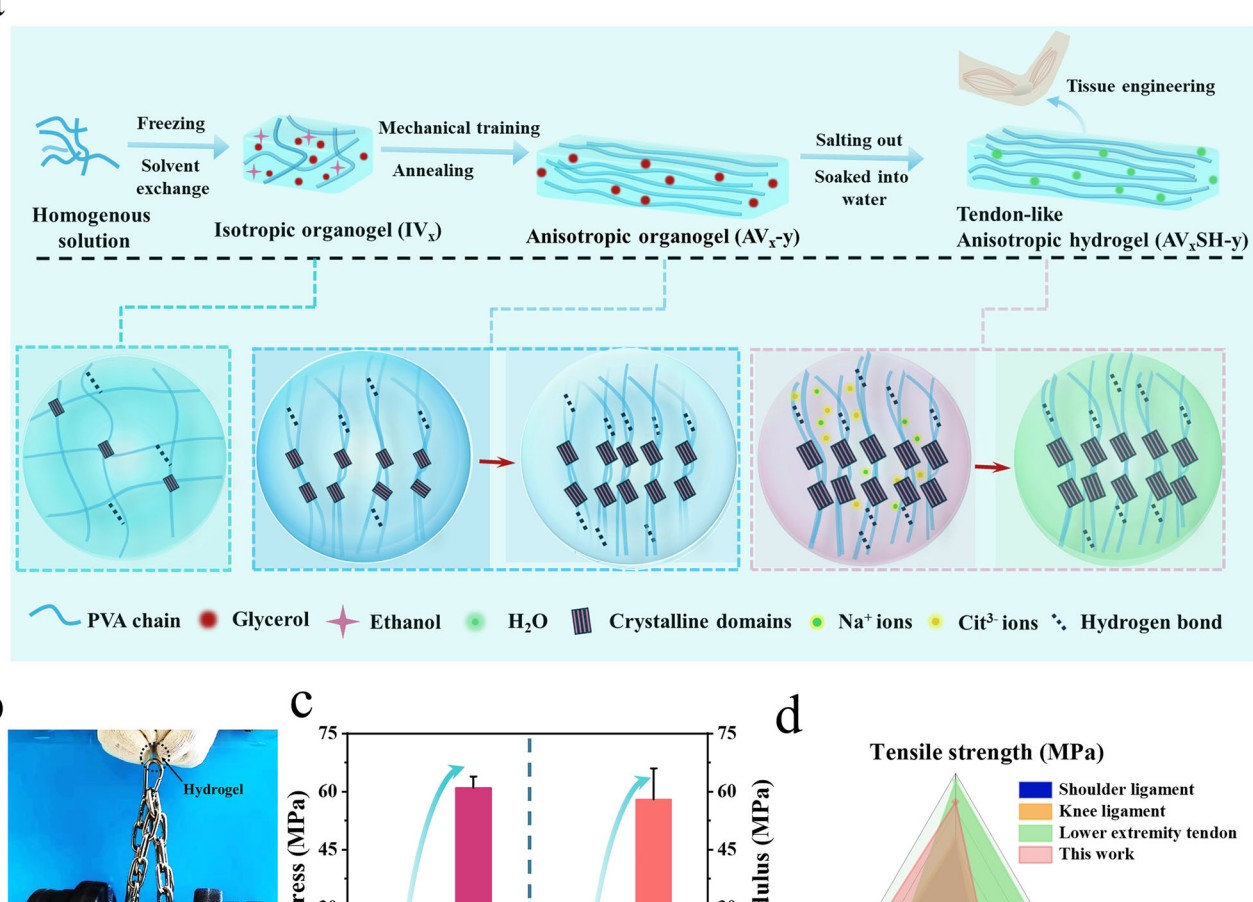

**Fig. 1 | Schematic illustration of the fabrication process and performance of anisotropic hydrogels. a** Preparation of anisotropic hydrogels via organogel mechanical training, wet annealing, and the salt-locking strategy. **b** Demonstration of the hydrogel's mechanical strength, as evidenced by a hydrogel strip lifting an 8 kg weight. **c** Comparison of tensile stress and Young's modulus between the pure (isotropic) hydrogel and the engineered anisotropic hydrogel. Data are presented as mean values ± SD, $n = 3$ independent samples. **d** Comparative evaluation of the mechanical properties of the developed hydrogel relative to various native biological tissues.

anisotropic organogels obtained through mechanical training; $AV_xH$ denotes anisotropic hydrogels obtained through mechanical training; $AV_xSH$ represents anisotropic hydrogels prepared by combining mechanical training and salting out; $AV_xS$-y specifies anisotropic organogels fabricated through mechanical training, wet-annealing, and salting out; $AV_xSH$-y specifies anisotropic hydrogels fabricated through mechanical training, wet-annealing, and salting out, where x indicates the initial PVA mass fraction, and y denotes the wet-annealing temperature. By systematically tuning the initial PVA concentration, wet-annealing temperature, and processing methods, the mechanical properties and water content of PVA hydrogels can be precisely controlled over a broad range.

Figure 2a–c illustrate the typical stress-strain curves and corresponding mechanical properties of various hydrogels. $IV_{15}SH$ exhibited a tensile strength of $1.8 \pm 0.3$ MPa, elongation at break of $771 \pm 53\%$, an elastic modulus of $0.4 \pm 0.1$ MPa, and a toughness of $6.0 \pm 0.5$ MJ·m$^{-3}$. Mechanical training induces macromolecular chain orientation in $AV_{15}SH$, amplifying strength (13.3 MPa, 7.4× increase), modulus (4.9 MPa, 11.1 × increase) and toughness

$(21.7 \pm 1.4$ MJ·m$^{-3}$, 3.6 × increase) while sacrificing stretchability to 385% due to restricted chain mobility. Notably, salt-out sample $(AV_{15}SH)$ outperforms non-salting out sample $(AV_{15}H)$ with a tensile strength of $5.2 \pm 0.6$ MPa, evidencing Hofmeister effect-driven hydrophobic self-assembly of aligned chains that amplify crystalline domain interconnectivity.

Wet annealing optimizes macromolecular conformations, eliminates internal defects, enhances intermolecular hydrogen bonding, promotes the formation of more crystalline domains, and increases the degree of crystallinity: elevating the annealing temperature from 90 °C to 120 °C enhances crystallinity, promotes the chain alignment and strengthens the polymer network, with tensile strength rising from $21.4 \pm 1.6$ MPa to $45.1 \pm 3.7$ MPa, elastic modulus increasing from $12.8 \pm 1.2$ MPa to $37.7 \pm 2.1$ MPa, and toughness improving from $30 \pm 2$ MJ·m$^{-3}$ to $66 \pm 12$ MJ·m$^{-3}$ for $AV_{15}SH$-90 and $AV_{15}SH$-120, respectively. This staged processing (alignment → conformational optimization →densification) with progressively improved crystallinity overcomes the strength-toughness trade-off inherent to conventional hydrogels, albeit at the cost of ductility.

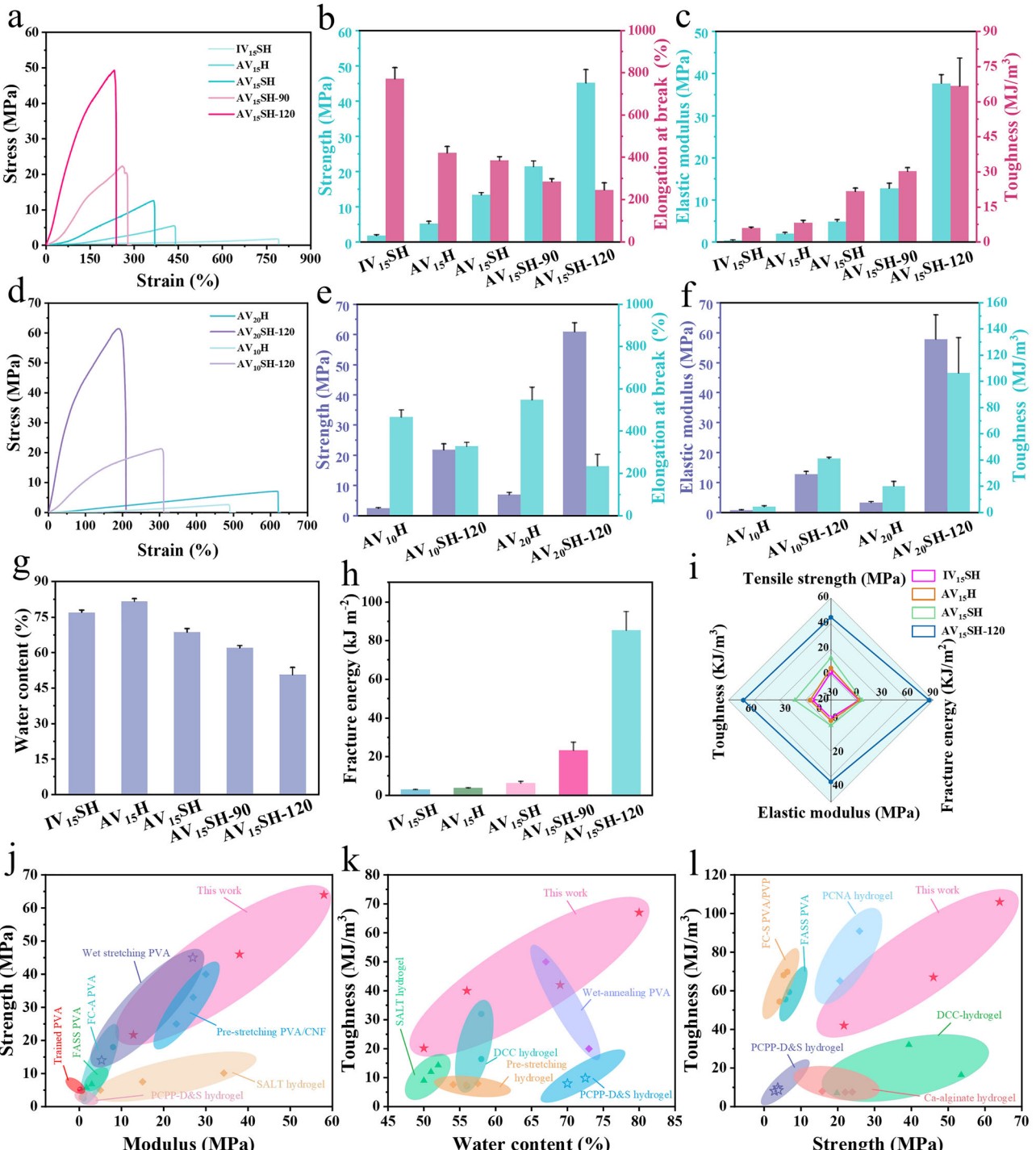

**Fig. 2 | Mechanical properties of PVA hydrogels. a** Tensile stress-strain curves of different PVA hydrogels (15 wt%), with a summary of their **b** tensile strength and fracture strain, and **c** modulus and toughness. **d** Tensile stress-strain curves of different PVA hydrogels (20 wt%), with a summary of their **e** tensile strength and fracture strain, and **f** modulus and toughness. Summary of **g** water content and **h** fracture energy of PVA hydrogels prepared using different methods. **i** Comparison of mechanical properties of $IV_{15}SH$, $AV_{15}H$, $AV_{15}SH$, and $AV_{15}SH$-90°. Comprehensive mechanical property comparisons: **j** strength vs. modulus, **k** toughness vs. water content, and **l** toughness vs. strength of the prepared PVA hydrogels alongside other tough hydrogels. All data are presented as mean values ± SD, $n = 3$ independent samples.

In this work, the stepwise preparation strategy was designed to regulate orientation and crystallization, thereby enhancing the mechanical performance of the final hydrogels. Figure S3 shows the mechanical properties of the gels at different intermediate states. After mechanical training, $AV_{15}$ exhibited a tensile strength of 44 ± 1 MPa, elongation at break of 220 ± 28%, elastic modulus of 43 ± 4 MPa, and toughness of 63 ± 11 MJ m$^{-3}$. Following

annealing, $AV_{15}$-120 demonstrated improved properties, with a tensile strength of 69 ± 2 MPa, elongation at break of 269 ± 48%, elastic modulus of 61 ± 10 MPa, and toughness of 131 ± 20 MJ m$^{-3}$. After salting-out, $AV_{15}S$-120 further improved to a tensile strength of 77 ± 2 MPa, elongation at break of 296 ± 17%, elastic modulus of 60 ± 4 MPa, and toughness of 156 ± 11 MJ m$^{-3}$. These results clearly demonstrate that the mechanical properties are progressively

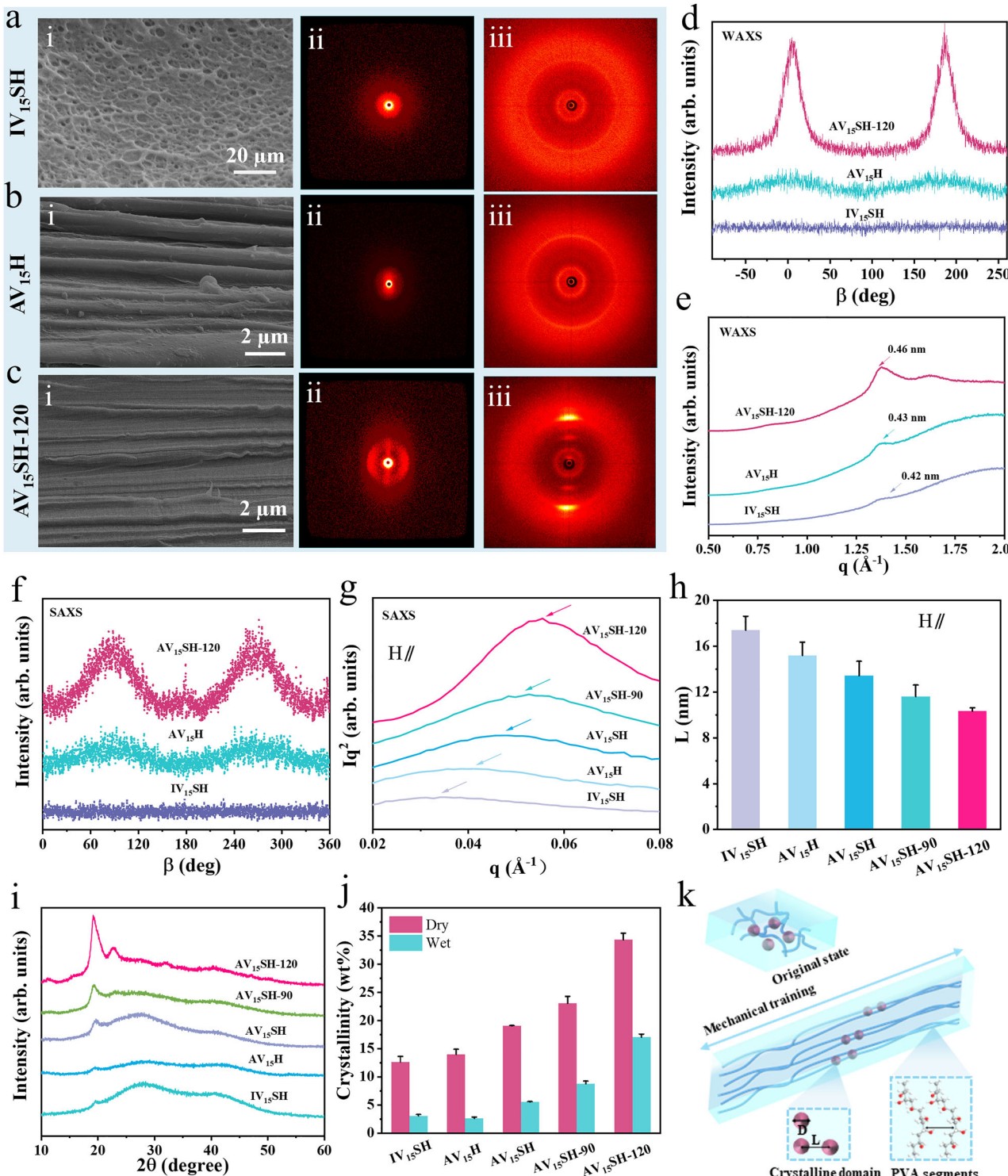

**Fig. 3 | Characterization of the micro-morphology and crystalline structure of different hydrogels.** Micro-morphology comparison of **a** IV$_{15}$SH, **b** AV$_{15}$H, and **c** AV$_{15}$SH-120, featuring: (i) SEM images, (ii) Small-angle X-ray scattering (SAXS) and (iii) wide-angle X-ray scattering (WAXS) patterns. **d** Azimuthally integrated intensity distribution of 2D WAXS patterns. **e** 1D WAXS curves showing scattering intensity vs. scattering vector (**q**). **f** Azimuthally integrated intensity distribution of 2D SAXS patterns. **g** 1D SAXS curves, depicting scattering intensity vs. scattering vector (q). **h** Calculated average distance between crystalline regions (All data are along the orientation direction of the nanofibers.). **i** X-ray diffraction (XRD) patterns. **j** Calculated crystallinity of hydrogels in dry and swollen states. **k** Schematic representation illustrating crystalline domain transformations. Crystallinity and L data are presented as mean values ± SD, $n = 3$ independent samples.

enhanced through the sequential steps of mechanical training, annealing, and salting-out.

We further investigated the influence of PVA concentration on the mechanical properties of hydrogels (Fig. 2d–f). As the PVA concentration increased from 10 to 20 wt.%, the tensile strength rose from $22 \pm 2$ MPa (AV$_{10}$SH-120) to $61 \pm 3$ MPa (AV$_{20}$SH-120), while the elastic modulus increased from $12.82 \pm 0.89$ MPa to $58 \pm 8$ MPa. Although a slight decrease in elongation at break was observed, AV$_{20}$SH-120 exhibited a maximum toughness of $106 \pm 27$ MJ·m$^{-3}$. However, AV$_{20}$SH-120 exhibited a high strength of $61 \pm 3$ MPa by

salting-out. Additionally, mechanical training, wet annealing, and salting out continued to enhance hydrogel properties across different PVA concentrations (Figs. S4 and S5). To clarify that the observed performance enhancements result from structural optimization rather than solely from concentration effects, we supplemented the analysis with normalized mechanical properties by polymer content. Figure S6a presents the unnormalized strength and modulus data, showing that both stepwise treatment and increased concentration contribute to improved mechanical performance, thereby necessitating a distinction between these two factors. The normalized results (Fig. S6b) demonstrate that, even after excluding the influence of polymer concentration, the strength and toughness of $AV_{20}SH$-120 are still 46-fold and 222-fold higher than those of $IV_{10}SH$, respectively. These findings confirm that the substantial performance enhancement originates primarily from the proposed structural optimization strategy.

The crack propagation resistance of materials is a critical indicator for evaluating their practical application potential. Since many practical applications require hydrogels to endure mechanical loads and resist failure under static or cyclic loading conditions, the ability to resist crack propagation is crucial. Therefore, we analyzed the fracture energy of different PVA hydrogels. As shown in Fig. 2h, all three treatments (mechanical training, wet annealing, and salting out) significantly improved the fracture energy of the hydrogels. Figure S7 presents the force–displacement curves of notched and unnotched samples. The fracture energy of $IV_{15}SH$ was calculated to be $3.1 \pm 0.1$ kJ·m$^{-2}$, whereas $AV_{15}SH$-120 exhibited a maximum fracture energy of $85 \pm 9$ kJ·m$^{-2}$, which is over 29 times higher than that of $IV_{15}SH$. The increase of fracture energy is mainly due to the synergistic effect of the anisotropic arrangement structure of the fiber bundles, the increase in crystallinity during wet annealing and the salting out effect, and the alignment of fibers perpendicular to the crack path effectively hindered crack propagation.

Notably, $AV_{15}SH$-120 surpasses the fracture energy of most biological tissues, including: skeletal muscle (~2.5 kJ·m$^{-2}$), cartilage (~1 kJ·m$^{-2}$) and tendons (~20–30 kJ·m$^{-2}$)[41]. Meanwhile, they possess strength comparable to that of natural tendons. Beyond their excellent mechanical performance, these anisotropic hydrogels also exhibit tunable water content (~50–90 wt%), closely matching that of biological tissues (Fig. 2g and Figure S8)[42].

Figure 2i compares the mechanical properties of $IV_{15}SH$, $AV_{15}H$, $AV_{15}SH$, and $AV_{15}SH$-120[23,30,31,34–45]. As expected, the anisotropic hydrogels demonstrated superior mechanical performance compared to $IV_{15}SH$, with $AV_{15}SH$-120 exhibiting notable advantages in tensile strength, elastic modulus, fracture energy, and toughness. Figure 2j–k provide a comparative analysis of state-of-the-art anisotropic hydrogels reported in previous studies[24,25,30,31,44–47]. Compared to the underwater mechanical training system, the hydrogels prepared by this strategy exhibit higher strength and broadly tunable water content[34]. Notably, the anisotropic hydrogels synthesized in this work achieved one of the highest tensile strengths and toughness values among hydrogels prepared using mechanical training, directional freezing, and pre-stretching methods. It is important to highlight that water content is a critical factor in evaluating hydrogels for biomedical applications.

Our progressive alignment mechanism creates a biomimetic strengthening and toughening architecture: tightly packed crystalline regions enhance strength via restricted chain mobility, and can resist crack initiation by the pining effect, while oriented amorphous regions enable viscoelastic energy absorption through reversible macromolecular chain slippage and hydrogen bond rupture. These findings establish a universal design rule: progressive alignment across molecular-to-mesoscopic scales enables hydrogels to bypass the strength-toughness trade-off, offering great potential for load-bearing biomedical applications.

## Characterization of the microstructures

A series of structural characterizations was conducted to reveal how the progressive reinforcement strategy (mechanical training → wet-annealing → salting-out) induces hierarchical alignment that bridges molecular ordering to macroscopic mechanical anisotropy. SEM, SAXS and WAXS were performed to examine their anisotropic structures. Scanning electron microscopy (Fig. 3a-i–c-i) demonstrates a structural evolution from isotropic pores (Freeze-dried $IV_{15}SH$) to tendon-mimetic aligned nanofibrils. $AV_{15}H$ exhibits an oriented fibrous structure along the training direction (Fig. 3b-i), whereas $AV_{15}H$-120 displays a denser and more highly aligned fibrous architecture (Figure S9), and the highly orientated structure is well maintained for $AV_{15}SH$-120 (Figs. 3c-i and S10), where confirmed that mechanical training initiates macromolecular orientation, wet-annealing enhances interchain hydrogen bonding and promotes macromolecular chain alignment, and salting-out locks the aligned configuration through crystallite stabilization. The anisotropic hydrogels exhibit a structure similar to that of tendons, making them promising for tissue engineering applications (Fig. S11).

To investigate the strengthening and toughening mechanisms of the anisotropic hydrogels, wide-angle X-ray scattering (WAXS) and small-angle X-ray scattering (SAXS) analyses were performed to examine their nanostructures. As shown in Fig. 3a-ii and a-iii, the 2D SAXS and WAXS patterns of $IV_{15}SH$ exhibit clear isotropy, with nearly uniform scattering intensities in all directions. However, the SAXS and WAXS patterns of $AV_{15}H$ and $AV_{15}SH$-120 display significant anisotropy, as indicated by the transformation of regular diffraction rings into diffraction arcs, confirming an oriented structure (Fig. 3b, c-ii, and b, c-iii). Moreover, the azimuthal integral intensity distribution curve of the WAXS model for $AV_{15}SH$-120 exhibits a sharper peak compared to $AV_{15}H$ (Fig. 3d), indicating an increased degree of molecular orientation. The orientation degree ($\Pi$) (Fig. S12), calculated from the azimuthal-integrated intensity distribution curves of the WAXS patterns, reveals that the high alignment of PVA molecular chains is primarily responsible for the anisotropic hydrogel's structure. Furthermore, $AV_{15}SH$-120 exhibits a higher orientation degree ($\Pi = 0.87$) than $AV_{15}H$ ($\Pi = 0.5$), confirming that the wet-annealing and salting-out process effectively promote and stabilize the oriented structure.

As shown in Fig. 3e, the strongest peaks in the 1D WAXS patterns of $IV_{15}SH$, $AV_{15}H$, and $AV_{15}SH$-120 appear at 0.42 nm, 0.43 nm, and 0.46 nm, respectively, corresponding to the average distance between adjacent PVA segments in the nanocrystalline microregions[48]. These results indicate that the progressive reinforcement strategy promotes the aggregation of PVA chains and hence causes nanocrystalline densification. Figure 3f presents the azimuthal integral curves of the 2D SAXS patterns, which exhibit a trend similar to that of the 2D WAXS azimuthal integral curves. This further demonstrates that mechanical training induces the oriented alignment of polymer molecular chains, forming an anisotropic, permanently locked structure parallel to the training direction due to salting-out effect. To quantify the evolution of crystalline domains during the progressive reinforcement process, the scattering intensity was corrected by multiplying it by the square of the scattering vector ($Iq^2$) (Fig. 3g). The raw, uncorrected 1D SAXS scattering curves (Parallel to the orientation direction) are shown in Fig. S13. 1D SAXS curves (Perpendicular to the orientation direction) are shown in Fig. S14. Additionally, using the Bragg equation ($L = 2\pi/q_{max}$), the average distance between adjacent crystalline domains is calculated. The results indicate a reduction in the parallel inter-domain distance from $17.40 \pm 1.21$ nm for $IV_{15}SH$ to $10.33 \pm 0.28$ nm for $AV_{15}SH$-120 (Fig. 3h), suggesting that the aligned PVA polymer chains and fibers become more closely packed due to the forces exerted during mechanical training and salting-out.

To offer more robust support for "stepwise construction" mechanism, structural characterization of intermediate structures

after each key processing step was performed. The 2D WAXS patterns of $AV_{15}$, $AV_{15}$-120, and $AV_{15}S$-120 consistently exhibited anisotropy with gradually intensifying diffraction arcs (Fig. S15a), confirming the formation and progressive enhancement of anisotropic structures during the stepwise fabrication process. The azimuthal intensity distribution profile of $AV_{15}S$-120 is noticeably sharper than that of $AV_{15}$, reflecting a higher degree of orientation induced by salting-out, which effectively locks the molecular alignment (Fig. S15b). The high orientation degree in the anisotropic gels is primarily derived from the mechanically induced alignment of PVA chains, while the salting-out step further stabilizes this oriented architecture. The orientation degree of $AV_{15}H$-120 was lower than that of $AV_{15}SH$-120, revealing the crucial role of salting-out in preserving chain alignment. In addition, the strongest peak in the WAXS 1D profile appears at $q = 1.57\,Å^{-1}$ (Fig. S15c), corresponding to the average interchain spacing between adjacent PVA chains in nanocrystalline regions, while the weaker peak at $q = 0.89\,Å^{-1}$ represents the average interlayer spacing within nanocrystalline domains; notably, the stepwise preparation exerts little influence on these two crystalline parameters. As shown in Fig. S16a, the 2D SAXS patterns of $AV_{15}$, $AV_{15}$-120, and $AV_{15}S$-120 exhibit clear anisotropy. In the parallel direction (chain alignment), the average inter-domain distance varies only slightly. In the perpendicular direction, the average inter-domain distance decreased from 9.39 nm to 8.36 nm and further to 6.30 nm (Fig. S16f), indicating that salting-out promotes structural densification of the gel. We summarized crystal dimension (D), inter-crystal spacing ($L_1$), inter-crystal spacing ($L_2$), inter-laminar spacing ($L_3$), inter-molecular spacing ($L_4$), and orientation degree of $AV_{15}$, $AV_{15}$-120, and $AV_{15}S$-120 in Table S1.

X-ray diffraction (XRD) was conducted to characterize the crystalline behavior of different PVA hydrogels. As shown in Fig. 3i, all PVA hydrogels have diffraction peaks around $2\theta = 19.7°$, corresponding to the (10$\bar{1}$) reflection plane of semi-crystalline PVA hydrogels[49]. The peak of $IV_{15}SH$ was small and broad with very low intensity, whereas $AV_{15}SH$-120 displays a distinct crystalline peak, indicating that the progressive reinforcement strategy enhances the crystallinity of PVA hydrogels. Additionally, $AV_{15}$, $AV_{15}$-120, and $AV_{15}S$-120 exhibited a diffraction peak near $2\theta = 19.7°$. The peak of $AV_{15}$ is small and broad with low intensity (Fig. S17a), whereas $AV_{15}S$-120 displays a distinct crystalline peak, indicating an increase in the crystallinity of the PVA gel. Furthermore, we calculated the average crystal size (D), which increased from 4.48 nm to 7.8 nm (Fig. S17b). After the annealing step, the hydrogel exhibited a small diffraction peak at $2\theta \approx 23°$, corresponding to the (200) crystal plane of the PVA crystalline structure[50], which was nearly undetectable in the non-annealed hydrogel. The emergence of this peak can be attributed to the enhanced chain mobility during wet annealing, which facilitates conformational rearrangement of macromolecules, elimination of structural defects, and promotion of crystallite growth. The salting-out step further strengthens intermolecular hydrogen bonding and promotes tighter chain packing, leading to an increase in crystallinity and higher intensity of this (200) reflection. Therefore, the appearance and enhancement of this small diffraction peak can be ascribed to the synergistic effects of wet annealing and salting-out. Furthermore, comparison among gels annealed at different temperatures reveals that higher annealing temperatures enhance the (200) peak intensity, consistent with the observed improvements in mechanical performance. To quantitatively assess crystallinity, differential scanning calorimetry (DSC) was used to analyze the crystalline behavior of hydrogels in the dry state, and the crystallinity in the swollen state was also calculated using Eq. 8. As shown in Fig. S18, all hydrogels exhibit clear endothermic peaks, with $AV_{15}SH$-120 displaying the most pronounced peak, suggesting the highest content of crystalline domains. Based on enthalpy calculations, the crystallinity in the dry state for $IV_{15}SH$, $AV_{15}H$, $AV_{15}SH$, $AV_{15}SH$-90, and $AV_{15}SH$-120 is $12.63 \pm 0.98\%$, $14 \pm 0.92\%$, $19 \pm 0.1\%$, $23 \pm 1.23\%$, and $34.3 \pm 1.17\%$, respectively, while the corresponding crystallinity in the swollen state

is $3.1 \pm 0.23\%$, $2.6 \pm 0.26\%$, $5.53 \pm 0.05\%$, $8.77 \pm 0.47\%$, and $17.03 \pm 0.55\%$, respectively (Fig. 3j). These results demonstrate that the crystallinity of anisotropic hydrogels prepared using the progressive reinforcement strategy is significantly enhanced due to the synergistic effects of mechanical training, annealing and salting-out.

As shown in Fig. S19, the average crystal dimension (D) was calculated using Scherrer's equation ($D = k\lambda / \beta cos\theta$)[51]. The estimated size of the crystalline domains in the hydrogels increased from 5.9 nm of $IV_{15}SH$ to 7.6 nm of $AV_{15}SH$-120. As illustrated in Fig. 3k, the average distance between adjacent crystalline domains decreases, while the crystal size increases, and the spacing between PVA molecular chains remains unchanged throughout the progressive reinforcement process. These results indicate that the enhancement of crystallinity is attributed to the densification of crystalline domains and the increase in crystal size. To comprehensively verify chain densification from the crystallization perspective, we have incorporated combined results from DSC, WAXS, SAXS, and XRD analyses (Table S2). DSC results reveal a gradual increase in crystallinity. XRD results show a progressive growth in crystal size across the anisotropic hydrogels. SAXS analysis indicates that the parallel inter-domain distance decreases significantly from 17.4 nm to 10.3 nm, whereas the perpendicular inter-domain distance decreases significantly from 18.4 nm to 8.5 nm, clearly demonstrating salting-out–induced chain aggregation and densification. Meanwhile, WAXS data show that the average chain spacing and interlayer distance remain essentially unchanged, thereby excluding their contribution. Taken together, these multi-technique results confirm that polymer chain densification primarily arises from macromolecular chain alignment, increased crystallinity and crystal size, and reduced inter-domain spacing.

FTIR spectroscopy elucidates how progressive reinforcement directs hydrogen-bond reorganization to drive hierarchical crystallization, a critical mechanism underlying the hydrogels' mechanical superiority. The systematic redshift of hydroxyl stretching vibrations ($3305\,cm^{-1}$ of $IV_{15}SH$ to $3276\,cm^{-1}$ of $AV_{15}SH$-120, Fig. S20) evidences intensified intermolecular hydrogen bonding, where mechanical training aligns PVA chains to optimize the interactions between hydrogen bonds, wet-annealing stabilizes these bonds through thermal relaxation, and salting-out induces $Na^+$-mediated dehydration to amplify bond density. Concurrently, the intensified C-O vibration at $1143\,cm^{-1}$ in $AV_{15}SH$-120 correlates with crystalline phase development, confirming that aligned hydrogen-bond networks template oriented crystallite growth[49]. This synergy creates a biomimetic "sacrificial bond" system: densely packed crystalline domains provide structural rigidity, while dynamically hydrogen bonds dissipate energy through reversible rupture-reformation cycles. Crucially, the strategy achieves a high swollen-state crystallinity (17.03%) by locking hydrogen-bond configurations against water plasticization, overcoming the classical crystallinity-swelling dichotomy in hydrogels. These findings establish hydrogen-bond engineering as a universal lever to tailor crystallization kinetics, enabling hydrogels to transcend strength-toughness-swelling trade-offs through molecular-scale network control.

## Crack-propagation resistance

Emerging applications of hydrogels in tissue engineering require them to remain stable under cyclic mechanical loading. While many tough hydrogels exhibit excellent mechanical properties under single loading, they often fracture under cyclic loading, limiting their practical applications[52]. Fatigue resistance, a critical material property that prevents crack propagation under cyclic mechanical stress, is essential for long-term durability[39,53,54]. Loading-unloading experiments at different applied strains (40–240%) were conducted to investigate the energy dissipation mechanism of hydrogels. As shown in Fig. S21, the loading-unloading curves of all hydrogels displayed distinct hysteresis loops, with the loop area increasing as strain increased. This trend

indicates progressive hydrogen bond breakage and enhanced energy dissipation during stretching. 1000 cyclic loading-unloading tensile tests at a constant strain of 100% for different hydrogels were performed to study the dissipated energy. As shown in Fig. S22, all samples exhibited significant mechanical hysteresis in the first cycle, known as the Mullins effect[55]. Notably, $AV_{15}SH$-120 demonstrated the largest hysteresis loop, with dissipated energy increasing from 7.47 kJ m$^{-3}$ for $IV_{15}SH$ to 1959 kJ m$^{-3}$ for $AV_{15}SH$-120 (Fig. S23). This exponential gain stems from synchronized hydrogen bond rupture-reformation and reversible chain slippage.

As discussed earlier, mechanical training, wet annealing, and salting out enhance hydrogen bonding and crystallinity, requiring greater energy dissipation to break hydrogen bonds and crystalline domains during cyclic stretching. The stress gradually decreased with an increasing number of cycles, and the loading-unloading curves stabilized after approximately 50 cycles. To further evaluate the fatigue resistance of hydrogels, single-notch tests underwater were conducted to determine crack extension resistance and fatigue thresholds. As shown in Fig. 4a, b, the fatigue thresholds of $IV_{15}SH$ and $AV_{15}H$ were relatively low (~87.2 J m$^{-2}$ and ~204.9 J m$^{-2}$, respectively) due to their low crystallinity. Following mechanical training and salting out, the fatigue threshold of $AV_{15}SH$ increased to ~731.3 J m$^{-2}$ (Fig. 4c). Additionally, wet-annealing treatment further enhanced the fatigue resistance of hydrogels. As shown in Fig. 4d, e, the fatigue thresholds of $AV_{15}SH$-90 and $AV_{15}SH$-120 increased to ~1259.6 J m$^{-2}$ and ~2083 J m$^{-2}$, respectively, which can be attributed to the substantial increase in crystallinity and hydrogen bonding. Figure 4f summarizes the fatigue thresholds of different hydrogels[56,57], illustrating the gradual increase in fatigue resistance following mechanical training, wet annealing, and salting out. To further evaluate the crack growth resistance of $AV_{15}SH$-120, a constant energy release rate of ~2083 J m$^{-2}$ was applied to notched samples for 10,000 cycles. No crack extension was observed along the notch direction during these cycles (Fig. 4g). Notably, after 5000 cycles, cracks branched along the loading direction rather than propagating along the notch, a hallmark of biological materials. This biomimetic failure mode redistributes energy through microcrack formation, preventing catastrophic failure[58]. Figure 4h presents the relationship between fatigue threshold and strength for different hydrogels[23,34]. The anisotropic hydrogels developed through the progressive reinforcement strategy exhibited both high strength and fatigue resistance, outperforming anisotropic hydrogels fabricated using other mechanical training methods reported in the literature. Figure 4i provides a schematic representation of the hydrogel's multiscale fatigue resistance mechanism. Aligned fibrils deflect cracks via anisotropic stress redistribution, reducing stress intensity at the crack tip. Crystalline domains act as rigid, highly functional cross-linkers, pin crack and impede crack growth. Besides, hydrogen bonds dissipate energy through reversible rupture-reformation. In summary, the synergy of anisotropic fibril alignment, crystalline-domain reinforcement, and dynamic hydrogen-bond networks enables stress redistribution, crack deflection, and efficient energy dissipation. These multiscale mechanisms account for the hydrogel's exceptional resistance to crack growth under cyclic loading and explain why cracks deflect along fibril bundles instead of propagating directly from the notch.

### Resistance to swelling and biocompatibility

Resistance to swelling is a critical property for hydrogels in biological applications. To assess this, hydrogels ($AV_{15}SH$-120) were immersed in deionized water, physiological saline, and PBS solution for 3 and 7 days. The hydrogels retained their original shape and size, maintaining structural integrity (Fig. S24). As shown in Fig. 5a, the swelling ratio remained low, consistently below 1.2%. This superior anti-swelling performance is attributed to the presence of dense crystalline domains that function as effective cross-linking points, restricting macromolecular expansion. Furthermore, the hydrogel was ultimately

obtained by swelling in water, allowing the macromolecular network to reach near-equilibrium and thereby preventing further re-swelling. The swelling resistance of $AV_{15}SH$-120 was further verified by tensile testing before and after immersion in deionized water for 7 days (Fig. 5b). The results demonstrated negligible changes in mechanical properties, with retention coefficients of tensile strength, fracture strain, and toughness reaching 90%, 85%, and 95%, respectively (Fig. 5c). These findings confirm the hydrogel's anti-swelling behavior and mechanical robustness.

Leveraging its pronounced anisotropic structure, exceptional fatigue resistance, and anti-swelling properties, we evaluated the biocompatibility of $AV_{15}SH$-120 and its capability to regulate directional cell growth. The cytotoxicity of the hydrogel was systematically evaluated. NIH-3T3 mouse embryonic fibroblasts were assessed using a Cell Counting Kit-8 (CCK-8) assay, along with live/dead staining to examine cytocompatibility. As shown in Fig. 5d, cells cultured on standard tissue culture plates served as controls, where live cells were stained green (calcein-AM) and dead cells were stained red (propidium iodide). The $AV_{15}SH$-120 group exhibited a minimal presence of dead cells, comparable to the control group. Quantitative analysis (Fig. 5e) revealed a cell viability exceeding 95% on days 1, 3, and 5, with a particularly high survival rate of 98.46% ± 2.72% on day 5, demonstrating no significant cytotoxicity. Regulatory effects of bioinspired hydrogels on cell growth: anisotropic hydrogels are known to regulate cell growth, migration, differentiation, and alignment by providing directional structural and mechanical cues[38,59]. The highly aligned fiber architecture and anisotropic mechanical properties of the hydrogel facilitated directed cell elongation. In contrast to the control group, where cells exhibited random growth, cells cultured on $AV_{15}SH$-120 demonstrated pronounced alignment along the fiber orientation (Fig. 5f). This suggests that the hydrogel's anisotropic structure provides biophysical cues for cell guidance, mimicking natural tissue architectures such as tendons and nerve fibers. Thus, the anisotropic hydrogels hold great promise for biofunctional applications, offering a platform for tissue engineering and regenerative medicine.

## Discussion

This study establishes an approach for fabricating anisotropic hydrogels through a bioinspired progressive reinforcement strategy (mechanical training → wet-annealing → salting-out), which achieves hierarchical alignment across molecular to macroscopic scales. Structural characterizations reveal that mechanical training induces macromolecular orientation, wet-annealing adjusts the macromolecular conformation, and salting-out locks aligned crystalline domains (34.3% crystallinity), achieving mechanical synergy: tensile strength (61 ± 3 MPa), toughness (106 ± 27 MJ·m$^{-3}$), and fatigue resistance (Γ = 2083 J·m$^{-2}$). The hydrogel's biomimetic architecture enables anisotropic energy dissipation: aligned fiber bundles and crystalline regions redistribute the stress and resist crack initiation, while dynamic hydrogen bonds dissipate cyclic stresses. Crucially, swollen-state crystallinity (17.03%) and salting-out-stabilized alignment ensure durability in physiological environments, overcoming the swelling-mechanics trade-off. The anisotropic hydrogels show biocompatibility and can facilitate directed cell growth. By mimicking tendon's hierarchical mineralization and crack-deflection mechanisms, this strategy redefines hydrogel design principles, offering great potential for load-bearing biomedical applications where simultaneous strength, fatigue resistance, and biocompatibility are imperative.

## Methods

### Materials and chemicals

Polyvinyl alcohol (PVA-1799, 98-99% hydrolyzed, Aladdin), glycerol (Aladdin), Ethanol absolute (AR), Sodium citrate (Na$_3$Cit·2H$_2$O, 98%),

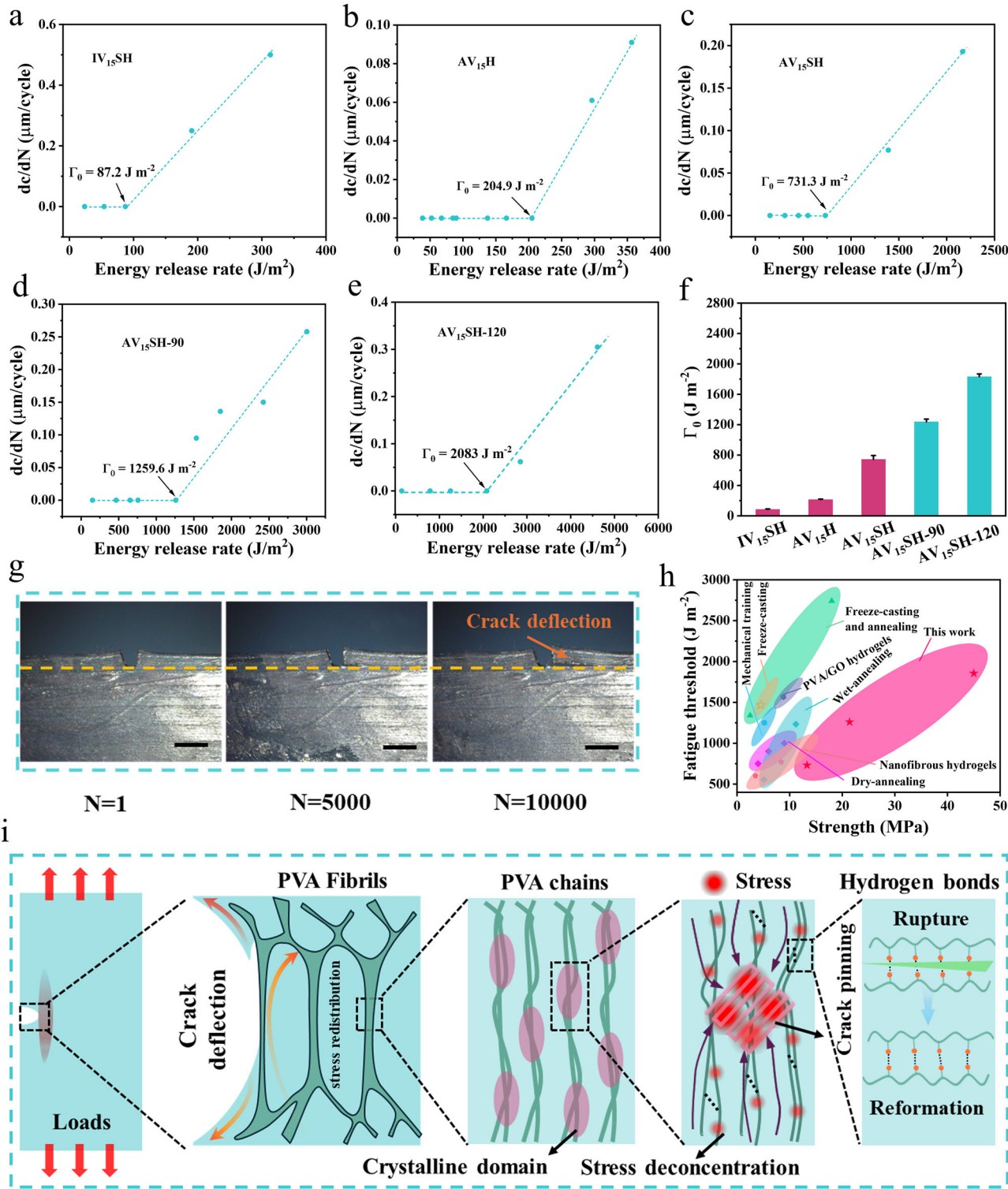

**Fig. 4 | Fatigue-resistance of different hydrogels.** Crack extension dc/dN per cycle versus applied energy release rate G of **a** $IV_{15}SH$, **b** $AV_{15}H$, **c** $AV_{15}SH$, **d** $AV_{15}SH$-90 and **e** $AV_{15}SH$-120. **f** Summary of fatigue thresholds ($\Gamma_0$) of different hydrogels. **g** Photographs of the notched sample at cycle numbers of 5000 and 10,000 for validation of the fatigue threshold as high as -1856 J m$^{-2}$ for $AV_{15}SH$-120. Fatigue thresholds data are presented as mean values ± SD, $n$ = 3 independent samples. **h** Diagram of fatigue threshold *versus* fracture strength for different hydrogels. **i** Schematic representation of the anti-fatigue mechanism of the anisotropic hydrogels.

glutaraldehyde (50 volume%, Aladdin), and hydrochloric acid (Sinopharm Chemical Co. Lit.) were used without further purification. All aqueous solutions were prepared using ultrapure deionized water.

## Fabrication of different PVA hydrogels
**Preparation of isotropic hydrogels.** PVA powder was dissolved in water and stirred at 100 °C for 5 h to obtain a homogeneous 15 wt% solution. The solution was then poured into polytetrafluoroethylene

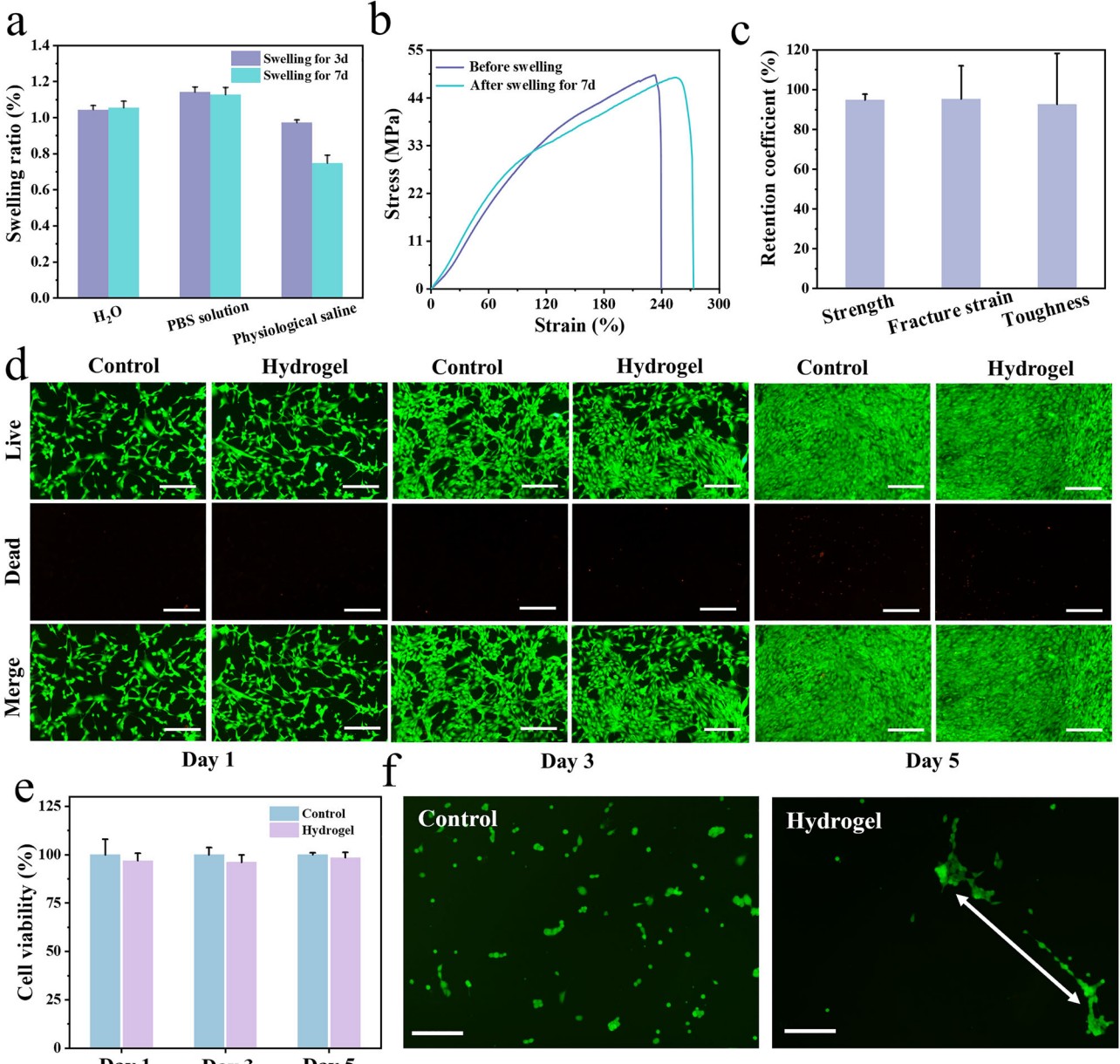

**Fig. 5 | Resistance to swelling and biocompatibility of AV$_{15}$SH-120. a** Swelling ratio of the hydrogel in different solutions. **b** Tensile stress-strain curves in water before and after 7 days of swelling. **c** Retention coefficient of mechanical properties after swelling in water for 7 days. Swelling ratio and retention coefficient data are presented as mean values ± SD, n = 3 independent samples. **d** LIVE/DEAD staining images of NIH−3T3 cells cultured on a standard cell culture plate (control) and AV$_{15}$SH-120 for 1, 3, and 5 days. Scale bars: 200 μm. **e** Proliferation ability of NIH−3T3 cells over 1, 3, and 5 days. At least three samples for each group were conducted and the results were represented as mean ± SD. **f** Fluorescence images of NIH-3T3 fibroblasts cultured on different substrates (*n* ≥ 20). Scale bars: 100 μm.

(PTFE) molds, cooled to room temperature, and subsequently frozen at −18 °C for 2 h. The resulting hydrogels were soaked in a glycerol/ ethanol solvent mixture for 48 h, thus obtaining the isotropic organogel IVx (where x indicates the initial PVA mass fraction). Some of the produced IVx specimens were processed to obtain a modified form of this intermediate. They were immersed in a 1.5 M sodium citrate solution for 12 h, followed by re-immersion in deionized water to yield PVA hydrogels, denoted as IV$_x$SH.

**Preparation of trained anisotropic gels.** The IV$_x$ organogels prepared as described in (1) underwent mechanical training for 30 cycles at a strain of 300% and a tensile rate of 200 mm/min. under ambient conditions, thus obtaining the 'trained' anisotropic organogel AVx (where x indicates the initial PVA mass fraction). Some of the produced

AVx specimens were processed to obtain a modified form of this intermediate. They were soaked in deionized water for 48 h, thus yielding the hydrogels AVxH.

**Preparation of trained and annealed anisotropic gels.** The organogels prepared as described in (2) were subjected to wet annealing at 120 °C for 30 min to obtain AV$_x$-y, where x indicates the initial PVA mass fraction, and y denotes the wet-annealing temperature. Some of the produced gels (AVx-y) were processed to obtain a modified form of this intermediate. They were soaked in deionized water for 48 h to obtain AVxH-y.

**Preparation of trained, annealed, and salted out anisotropic gels.** The annealed and trained anisotropic organogels AVx-y prepared as

described in (3) were immersed in a sodium citrate solution for 12 h to obtain AVxS-y, followed by soaking in water for another 48 h, which yielded the final product AVxSH-y, where x denotes the initial PVA mass fraction, and y denotes the wet-annealing temperature.

## Tensile test

All tensile tests of the hydrogels with a dumbbell shape were conducted on a universal tensile machine (Instron Model 3367, USA) at room temperature with a speed of 50 mm/min. The thickness of the samples was measured using a digital measuring device and the width was measured using a vernier calipers. The area under the stress-strain curves to failure is defined as the "toughness"[57]. The loading-unloading experiments under different applied strains (40–240%) were conducted at room temperature with a speed of 50 mm/min. A cyclic loading-unloading test at a fixed strain of 100% was conducted underwater to study the dissipated energy of the hydrogels. All mechanical tests were performed parallel to the nanofiber alignment.

## Pure shear tests

The universal testing machine (Instron Model 3367, USA) was used to test the tearing energy of the hydrogel, with a set tensile rate of 50 mm/min. Notched and un-notched hydrogels were used for the pure shear tests to measure the tearing energy. The fracture energy can be calculated by[50]:

$$\Gamma = \frac{W(\Delta L_c)}{A} \tag{1}$$

where $A$ denotes the cross-sectional area of the un-notched specimen, while $\Delta L_c$ represents the horizontal coordinate corresponding to the highest point on the force-displacement curve of the notched specimen.

## Fatigue tests

A single-notch method to determine the fatigue resistance of the hydrogels. To prevent dehydration of the hydrogel, all fatigue tests were conducted in a water bath with the initial crack length($c_0$) of the hydrogel, less than one-fifth of the specimen width ($L_0$). The hydrogels were performed using a customized mechanical stretcher (FULETEST, China) equipped with a 100 N loading cell. A digital camera (AF4915ZTL, Dino-Lite) was used to record the initial crack propagation ver continuous stretching cycles without relaxation. The curves of nominal stress S versus stretch $\lambda$ of the unnotched samples were obtained over Nth cycles with the maximum applied stretch of $\lambda_{max}$. By applying the same tensile strain ($\lambda$) to the unnotched sample, the $N^{th}$ cyclic strain energy, denoted as $W$, is calculated by:

$$W(\lambda_{max}, N) = \int_1^{\lambda_{max}} S d\lambda \tag{2}$$

The energy release rate ($G$) is calculated by:

$$G(\lambda_{max}, N) = 2k(\lambda_{max}) \cdot c(N) \cdot W(\lambda_{max}, N) \tag{3}$$

where k is a slowly varying function of the applied stretch as $k = 3/\sqrt{\lambda_{max}}$, c and W refer to the crack length and the integral area of the Nth loading part. The fatigue threshold was linearly extrapolated, and the anisotropic hydrogel was fatigued for 10,000 cycles without crack longitudinal extension.

## Water content measurement

The hydrogel was dried at 37 °C until a constant weight, and the initial and dried mass of the hydrogel were expressed as $m_a$ and $m_b$, respectively. Accordingly, the water content can be calculated by:

$$\phi_{water} = \frac{m_a - m_b}{m_a} \times 100\% \tag{4}$$

## SEM characterization

The hydrogel was first frozen in liquid nitrogen and then fractured by external force. Afterwards, it was subjected to freeze-drying using a SCIENTZ-10N freeze dryer. Prior to observation with a scanning electron microscope (Zeiss Supra55, Germany), the fracture surface of the freeze-dried hydrogel was treated with gold sputtering.

## WAXS and SAXS measurement

Wide-angle X-ray scattering (WAXS) and small-angle X-ray scattering (SAXS) measurements were conducted using a NanoSTAR instrument (Bruker AXS). The X-ray source was set with a wavelength of 0.154 nm, an operating voltage of 50 kV, and a current of 0.6 mA. For the WAXS analysis specifically, the diffraction profiles were recorded within a 2θ angular range spanning from 3° to 35°. The detector was positioned 1045 mm away from the sample, while the scattering vector (q) was adjusted to cover a range of 0.007 to 0.123 Å$^{-1}$. Additionally, the exposure duration for the measurement was set to 300 s. The intermolecular spacing, intercrystal spacing, and interlamellar spacing of the hydrogels were calculated as[48]:

$$L = \frac{2\pi}{q_{max}} \tag{5}$$

Orientation (∏) can be calculated as[60]:

$$\prod = \frac{180 - FWHM}{180} \tag{6}$$

where full width at half maximum (*FWHM*) is the width at half-maximum of the azimuthal distribution curve along the equatorial reflection.

## Measurement of DSC

Different samples were tested by differential scanning calorimetry (DSC 8500, Perkin Elme, USA). DSC tests were performed in a nitrogen atmosphere with a flow rate of 30 ml min$^{-1}$. The crystallinity of the dried sample ($X_{dry}$) is calculated as:

$$X_{dry} = \frac{H_{crystalline}}{H_{crystalline}^0} \times 100\% \tag{7}$$

where $H_{crystalline}^0$ (=138.6 J g$^{-1}$) is the enthalpy for fusing the PVA with 100 wt.% crystallinity at the equilibrium melting point $T_m^0$ [61]. The crystallinity in the swollen state ($X_{swollen}$) is calculated as:

$$X_{swollen} = X_{dry}(1 - \phi_{water}) \times 100\% \tag{8}$$

where $\phi_{water}$ (%) is the water content of hydrogels.

## X-ray diffraction characterization (XRD)

Regular X-ray diffraction (XRD-7000, 473 SHIMADZU, Japan) characterization was performed to evaluate the crystallinity behavior of hydrogel samples. The average size D of the crystal domains can be calculated using the Debye-Scherrer formula[50]:

$$D = \frac{k\lambda}{\beta \cos \theta} \tag{9}$$

where the dimensionless shape factor (k) is set as 1 if the shape of the crystalline domain is approximated as a sphere, $\lambda$ is the wavelength of x-ray diffraction, $\theta$ is the Bragg angle, and $\beta$ is the full width at half maximum of the peak.

### ATR-FTIR characterization
ATR-FTIR spectra were obtained using an infrared spectrometer (Cary610/670, Varian, USA) to characterize the vibrations of hydrogels (wet state) functional groups.

### Transmittance characterization
UV-visible-near infrared absorption spectrometer (Carry 5000, Varian, USA) with wavelengths of 400-800 nm was used to determine the visible light transmittance of the hydrogels.

### Cytotoxicity assays
The cytotoxicity of the hydrogel was qualitatively assessed using the CCK-8 method. A 300 μL suspension of NIH/3T3 cells (The cells were purchased from Suzhou Meilun Biotechnology Co., Ltd., with the cell catalog number PWE-MU002.) and the hydrogel were co-cultured in a 48-well plate for 1, 3, and 5 days, respectively. The medium was then replaced with CCK-8 solution and incubated for an additional 0.75 h. The absorbance (OD) of the supernatant at 450 nm was measured using a microplate reader (Model 550, Bio-Rad, Hercules, CA, USA), and the proliferation ability of NIH/3T3 cells was calculated by:

$$\text{Cell viability} = \frac{OD_{\text{sample}} - OD_{\text{blank}}}{OD_{\text{control}} - OD_{\text{blank}}} \times 100\% \qquad (10)$$

The LIVE/DEAD staining assay was further used to evaluate the cytotoxicity of the hydrogel. 1.0 mL suspension of NIH/3T3 cells (density = $5.0 \times 10^4$ cells/mL) was seeded into 6-well plates and co-cultured with the hydrogel for 1, 3, and 5 days, respectively. The changes in cell morphology under these conditions were observed using the LIVE/DEAD staining method. Finally, the cells were stained with a mixed dye solution containing 1.0 μM calcein-AM and 1.0 μM propidium iodide (PI), and image acquisition was performed using an inverted fluorescence microscope.

### Cell fluorescent staining
The ability of hydrogels to promote the oriented growth of cells was evaluated using the fluorescent staining method. 1.0 mL of NIH/3T3 cells were seeded into a 6-well plate and co-cultured with the hydrogels for 24 h. Subsequently, the cells were stained with a mixed solution containing 1.0 μM Calcein-AM and 1.0 μM propidium iodide (PI), and images were captured under an inverted fluorescence microscope.

### Reporting summary
Further information on research design is available in the Nature Portfolio Reporting Summary linked to this article.

## Data availability
Source data are provided with the paper. All data are available from the corresponding author upon request. Source data are provided with this paper.

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

## Acknowledgements

This study was financially supported by the National Natural Science Foundation of China (52473049), Jiangsu Provincial International Joint Laboratory of Optic/Electronic/Magnetic Functional Materials and Sensors, Qing Lan Project of Yangzhou University, and Jiangsu Province.

## Author contributions

J.G. and H.L. conceptualized the study and designed the experiments. H.L. performed the experiments. H.W., W.H., and C.G. assisted in completing part of the experimental testing. H.L. analyzed data and wrote the paper. J.G., H.L., W.H., C.G., W.S., and D.C. substantively revised the final version of the manuscript.

## Competing interests

The authors declare no competing interests.
