## [Transparent Peer Review file · Nature Communications]

Hierarchical Crack-Resistant, Tissue-Mimetic Hydrogels Enabled by Progressive Nanocrystallization of Anisotropic Polymer Networks

Corresponding Author: Professor Jie Feng Gao

Version 0:

Reviewer comments:

Reviewer #1

(Remarks to the Author)

Li and coworkers have developed a crack-resistant and tissue-mimetic hydrogel via a phase-transition-guided hierarchical engineering strategy that progressively constructs anisotropic polyvinyl alcohol networks. While previous studies have reported anisotropic PVA hydrogels, this work stands out by achieving a simultaneous and significant enhancement of multiple critical mechanical properties including strength, toughness, fatigue resistance, and notably, high stretchability (larger than 200% fracture strain with a tensile strength of ~60 MPa) by combining mechanical training, wet-annealing, and salting-out processes. I recommend this manuscript for publication after minor revisions, as it makes a meaningful and original contribution to the design of hierarchical hydrogels with improved mechanical and anti-swelling properties.

1: As known, the stretching method was often used to prepare anisotropic PVA hydrogels, and mechanical training of hydrogels in water was also reported to achieve the alignment of PVA macromolecules, and what are the unique advantages of the mechanical training of organogels and subsequently annealing in this study?

2: In many cases, achieving macromolecular alignment often compromises stretchability, and high strength and modulus typically result in low fracture strains (less than 100% for many strong anisotropic hydrogels). Interestingly, however, the PVA hydrogels in this work demonstrate a high strength of 61 ± 2.9 MPa while maintaining a large fracture strain exceeding 200%. Please provide a discussion on the underlying mechanisms and the significance of this mechanical advantage.

3: The title emphasizes the role of progressive nanocrystallization; however, the crystallization process is primarily introduced during the wet-annealing step (as shown in Fig. 1a). Please include a detailed description of how the crystalline structure evolves throughout the different stages of the fabrication process, and this will better elucidate the relationship between the processing stages and the development of the crystalline features that contribute to the hydrogel's enhanced mechanical properties.

4: In Fig. 2, the annealing temperature significantly influences the final mechanical properties of the hydrogels. Notably, AV15SH-120 exhibited markedly higher tensile strength and fracture energy compared to AV15SH-90. Could you elaborate on how the optimal annealing temperature was determined? Specifically, what criteria or experimental considerations guided the selection of 120°C as the optimal temperature? Additionally, did the higher annealing temperature risk damaging or compromising the anisotropic structures within the hydrogels?

5: In Fig. 3b and 3c, an apparent orientation was observed in both AV15SH-120 and AV15H hydrogels. Please explain the underlying reasons for the significant difference in the degree of orientation between these two hydrogels?

6: In Fig. 4g, the crack did not propagate in the 5000th cycle, and why did it propagate along the fibril direction (crack deflection) rather than the notch direction? Please also explain why and how the stress was concentrated in the crystalline domains (Fig. 4i).

7: As shown in Fig. 5, the hydrogel exhibited excellent swelling resistance, and how did the anisotropic structure as well as the crystalline domains affect this performance? Additionally, do the anisotropic hydrogels exhibit advantages in swelling resistance compared to the isotropic counterparts?

Reviewer #2

(Remarks to the Author)

[Editorial Note: Please see the end of the file]

Reviewer #3

(Remarks to the Author)

In their submitted Manuscript, the authors present very valuable results, which on the Reviewer's opinion are of considerable interest for the readers of Nature Communications. The work surely will achieve a prominent status in the fields of biomimetic materials, of advanced anisotropic hydrogels, and of materials design.

In the present study, the authors obtained bio-mimetic (tendon-like) hydrogels which display exceptionally good mechanical properties in their equilibrium-swollen state (and are resistant to any further swelling). In their gel preparation design, the authors were able to overcome the trade-off strength vs. toughness, and also swelling vs. mechanical properties, which is a great success.

The obtained hydrogels markedly surpass previously developed anisotropic gels of similar type, described in literature, and they even surpass advanced biological tissues in some critical properties.

A notable success consists in the achieved excellent fatigue resistance of the studied hydrogels, which is related to intrinsic mechanisms of crack pinning and deflection, as well as to energy dissipation via disruption/recombination of sacrificial hydrogen bonds.

The characteristics of the gels (including biocompatibility tests) suggest a great potential in biomedical applications, but possibly also in technical ones, such as soft robotics.

Of considerable own scientific value is the preparation procedure itself, which in a hierarchical manner builds-up a tendon-like anisotropic structure, and thus the mechanical strength and toughness of the hydrogels, in the sequence: Crosslinking of polyvinyl alcohol by initial partial crystallization via freeze drying, mechanical training by cyclic stretching in the presence of a poor plasticizer solvent, further crystallization and re-organization via annealing in the same solvent, solvent exchange for water, followed finally by an enhancement of hydrogen bonding via a salting-out treatment.

The obtained products display unusually attractive properties, and were well-characterized. The processing strategy which yielded the gels is very sound, and is of great scientific value for developing new advanced bio-mimetic materials in the future. The paper is well-written, meets high scientific standards, and all its conclusions are well-supported by data. The Experimental Section also is well-written and provides sufficient information for work reproduction in general, albeit some improvements are suggested below.

For all the above reasons, the Reviewer highly recommends the Manuscript for publication in Nature Communications.

Below are listed some minor technical issues, which the Reviewer recommends to be addressed (this can be done quickly), in order to achieve the maximum impact on the reader, and optimal information:

1) Experimental Section, lines 504–508:

The current description of gel annealing appears confusing, also in light of later discussion of gel synthesis and characterization in the Manuscript: After obtaining AV15SH (line 504) was the material really annealed as aqueous gel, immersed again in sodium citrate and finally soaked again in water (as it seems to be stated)? All the later discussion suggests that the authors inserted annealing as a new process step (after training, but before citrate treatment), in order to obtain the product series AV15SH-x. The most reader-friendly solution would be to introduce a new paragraph "(3) Preparation of annealed anisotropic hydrogels" for describing the preparation of AV15SH-x.

2) Introduction:

line 39: "Additionally, hydrogels exhibit excellent biocompatibility, ..."

and line 42: "However, their inherently nonuniform polymeric network ..."

suggested would be more precise comments, e.g.

line 39  "Additionally, SOME hydrogels ... etc. ..."

and line 42  "However, their OFTEN inherently ... etc. ..."

3) 2.1. Design of anisotropic hydrogels: line 100, Figure 1:

Figure 1c: suggested is some more distinct separation of the data referring to Modulus and to Stress, maybe a vertical line dividing the graph;

Figure 1d is a very intuitive one, but a version of better quality (resolution) or with higher contrast would be useful for the final version of the paper.

4) (2.1. Design of anisotropic hydrogels): adding a few words would be useful at the following lines:

line 114/115:

“Subsequent mechanical training under controlled ethanol volatilization ...”

a more specific description would be of interest for the reader at this place:  e.g.

“Subsequent mechanical training (30 cycles of stretching and relaxation up to 300%) under ... etc. ...”;

how was the volatilization controlled?

was there some significant creep during the training?

line 117:

“Wet annealing enables ...”  e.g. “Wet annealing IN GLYCEROL AT 90 OR AT 120°C enables ...”

line 119/120:

“... during salting-out, where ...”  e.g. “... during salting-out WITH SODIUM CITRATE, where ...”

5) (2.1. Design of anisotropic hydrogels):

line 128 (appearance of the final product): “... yielding transparent hydrogels (Figure S1, ...”:

a photograph of the gel would be desirable rather in Figure 1 (line 100) than in Figure S1; with optimal illumination, very impressive images of highly transparent gels (like the studied ones) can be obtained.

6) (2.1. Design of anisotropic hydrogels):

“(orientation factor >0.6) for ...”

suggested:  “(orientation factor >0.6, as observed by WAXS, SAXS) for ...”

7) 2.2. Mechanical properties: discussion text (lines 151-225):

It would be useful to describe the family of the studied products, including the meaning of codes like “IV15SH”, etc., either at the begin of this chapter, or at the end of the previous one (2.1. Design of anisotropic hydrogels).

8) 2.2. Mechanical properties: discussion text (lines 151-225):

The reviewer would recommend to split the discussion into more paragraphs, and to arrange them in a sequence following the steps of the gels preparation (and their mechanical/structure consequences), plus commenting additional aspects like: PVA concentration, Resistance to crack propagation, Comparison with biological tissues, Comparison with competing mechanically trained systems.

In the present text, new paragraph breaks would be suggested at lines 165, 167, 191, 203 and 212, in order to highlight their topics (ideally, the first words in each paragraph should give a quick orientation).

9) 2.3. Characterization of the microstructures:

line 257: “... evolution from isotropic pores (IV15SH) to ...”

suggested:  “... evolution ... etc. ... pores (FREEZE-DRIED IV15SH) to ...”

10) (2.3. Characterization of the microstructures):

line 304/305: “... swollen state was also calculated.”

suggested addition of short explanation:  “... swollen ... etc. ... calculated (USING SI-EQUATION 8, TAKING INTO ACCOUNT THE ABSORBED SWELLING WATER AS CONTRIBUTION TO THE AMORPHOUS PHASE).”

-related Fig. S.11 in SI file: a more descriptive scaling of the Y-axis would be from 0 to 8.0 (rather than 5.6 to 8.0).

11) line 323 (FTIR): it would be useful to specify, whether the FTIR spectra were recorded in dry or in the wet state.

12) paragraph breaks would be suggested (as topic separation) at lines: 256, 285, 289, 297, and 302 (ideally, the first words in each paragraph should give a quick orientation).

13) 2.4. Crack-propagation resistance:

line 344: Figure 4: possibly one representative graph (from the collection a–e) from Fig. S12 and one from Fig. S13 should be added to the composite Figure 4, because these cyclic characteristics are really interesting;

line 369: it would be interesting to add an eventual short comment comparing the different creep tendencies of the samples in Fig S13.

14) The cyclic tests illustrated by Fig. S12 and Fig. S13 should be described in the Experimental Section of the Supporting Information File, as information about them is missing;

-were these tests conducted in H₂O (‘underwater’)?

15) paragraph breaks would be suggested (as topic separation) at lines 358, 363, and 374 (ideally, the first words in each paragraph should give a quick orientation).

16) 2.5. Anti-swelling performance and biocompatibility.

suggested: “Resistance to swelling” instead of “Anti-swelling performance” (or Anti-swelling properties/behavior, etc.).

17) paragraph break would be suggested (as topic separation) at line 456, concerned with cell growth (ideally, the first words in the paragraph should give a quick orientation).

Version 1:

Reviewer comments:

Reviewer #1

(Remarks to the Author)

The author has made revisions to the article based on the suggestions, and the article can be accepted.

Reviewer #2

(Remarks to the Author)

In response to my earlier comments, the authors have made a number of useful revisions that strengthen the manuscript. They updated the process diagram to show the annealed state separately and explained more clearly that densification happens during salting-out, while the alignment is largely kept after soaking. They also added many results of intermediate samples to support this. In addition, they provided mechanical data normalized by polymer content, showing that the performance is still much better even when the effect of concentration is excluded (for example, the strength and toughness of AV20SH-120 are still far higher than IV10SH on a per-content basis). They corrected the orientation labels and added raw 1D SAXS curves in the SI for transparency. Overall, the revised paper is much stronger, and the authors now explain more clearly that combining organogel mechanical training, wet annealing, and citrate salting-out gives anisotropic, crystallinity-reinforced PVA hydrogels with high fatigue threshold and a rare balance of strength, toughness, and swelling resistance. However, there are still the following detailed issues that require further improvement.

1. The manuscript contains too much supplementary information placed within parentheses, which interrupts the flow. Please consider streamlining by reducing the parenthetical content or integrating it into the main sentences.
2. In Figure S6, the modulus data of IV₁₀SH appear to be missing. If this is due to the values being too small to be clearly displayed, I recommend adding an enlarged inset bar chart of IV₁₀SH strength and modulus in the blank space of this figure to ensure data completeness and readability.
3. Please carefully check figure-level details for consistency and legibility. For example: Fig. 1(c) bars have outlines whereas other bar charts do not (style inconsistency); the central region of Fig. 2(i) is not readable; Fig. 4(f) bar chart shows black striping artifacts; error bar styles are inconsistent in Fig. S6 and Fig. S16(f); the numeric labels below Fig. S12 do not align with tick marks (please unify with the formatting used in Fig. S17(b)); in Fig. S23(a, d) the text color on the right side appears incorrect. Some of these may stem from source image quality; in any case, please re-export/replace assets and unify styles (e.g., bar fill without borders, consistent error-bar caps, tick/label alignment, adequate resolution or vector format).

Reviewer #3

(Remarks to the Author)

[Editorial Note: Please also see the attachment at the end of the file]

The reviewer 3 is very impressed by the results presented by the authors (as stated previously). The Authors also responded to all the points raised by the reviewer 3, and they also added interesting and valuable additional data and discussion, while responding to the reviewers 1 and 2.

After reading the revised versions of the Manuscript and SI file,

The reviewer 3 strongly suggests one correction (Figure 1a and Experimental: Preparation description) which is specified in an attached File in MS Word, which is important for the readers orientation in the family of the studied products and intermediates.

For the authors consideration, the reviewer 3 also suggests two eventual short comments to be possibly added to the discussion (points 2 and 3).

Technical: in some cases, the presented values and standard deviations should be rounded, e.g. "268.67 ± 48.22%"  "270 ± 50%", or at least "269 ± 48%".

1) Improvement of

Figure 1a

is strongly suggested,

in order to display all the discussed products and intermediates, as outlined graphically in the MS Word file attached by the reviewer 3.

The related

Experimental description

in the Main Manuscript

also should be improved as suggested in the mentioned File attached by the reviewer 3.

The suggested corrections only modestly alter the actual image and the experimental description, while the reader will obtain a good orientation concerning the discussed products and intermediates, especially in view of the newly added discussion of some of the interesting intermediates.

2) Discussion of importance, sequence and synergy of the gel modification steps:

(lines 93-101):

Here, in the revised Discussion, the authors very now very well describe the synergy and the sequence of the applied modification steps.

For the authors' consideration:

> It also might be useful to point out that, the employed modification steps were applied to the gels in the sequence of their decreasing 'aggressivity'. In spite of the mildness of the (last) salting-out step, its effect on the mechanical properties (tensile tests), as well as on the crystallinity (XRD) was very pronounced. The kosmotropic and multiply hydrogen-bonding citrate anion appears to be very helpful in this final reorganization step where it causes additional crystallization and favors a defect-free structure (witnessed by XRD: the relatively strong 200 reflection at 22.3deg).

> The gradual ordering of the structure of the anisotropic hydrogels crosslinked by hydrogen bonded nanocrystallite domains could be commented in analogy to the comparison of irregular covalently crosslinked hydrogels (divinyl co-monomers, free radical polymerization) vs. hydrogels with an ideally homogeneous network structure (achieved by special synthesis routes). The regular network structure leads to much improved tensile properties, especially toughness. In the studied case, the physical crosslinks in the IVx gels are rather randomly distributed, similarly like the network junctions in the irregular covalently crosslinked hydrogels. Literature examples about the covalent hydrogels include [Sakai et al., *Macromolecules*, 2008, 41, 5379–5384. DOI: 10.1021/ma800476x] for the ideally homogeneous ones, or e.g. [Dusek et al., *Polymer Bulletin*, 1980, 3, 19–25. DOI: 10.1007/BF00263201], [Shibayama, *Macromol. Chem. Phys.* 1998, 199, 1–30. DOI: [https://doi.org/10.1002/\(SICI\)1521-3935\(19980101\)199:1<::AID-MACP1>3.0.CO;2-M](https://doi.org/10.1002/(SICI)1521-3935(19980101)199:1<::AID-MACP1>3.0.CO;2-M)] for the irregular ones.

3) Tensile properties: Figure 2 a, d:

A comment might be useful (no additional graphs or experiments) about eventual elastic and plastic regions (deformation limits) of the tensile curves, in relation to high-slope low and slope regions in the tensile graphs.

In this context: How large is the visually observed approximate creep value after sample disruption (if the pieces are put together; the results in Fig. S 21 might suggest some creep)? E.g. is the lower-slope region at the high elongation values predominantly plastic (e.g. disconnection or reorganization of less perfect crystallites)? The relation of the individual preparation steps to the eventual elastic / plastic deformation regions might be interesting.

Dr. Adam Strachota, PhD

Dear Editors and Reviewers:

Thank you for your letter and for the reviewers' comments concerning our manuscript entitled "Hierarchical Crack-Resistant, Tissue-Mimetic Hydrogels Enabled by Progressive Nanocrystallization of Anisotropic Polymer Networks" (Manuscript ID: NCOMMS-25-30700A). We have incorporated the suggestions and comments in the revised manuscript, and supplemented additional experiments, particularly regarding the WAXS and SAXS characterization of the transitional gel states. These changes have been highlighted in red color. For easy reference, point-by-point responses to the comments of the reviewers are also given below in red color.

We hope we have been able to address the reviewer's concerns, and we would like to thank the reviewers again for taking the time to provide their helpful input throughout the entirety of this review process.

Response to reviewers:***Reviewer #1 (Remarks to the Author):***

Li and coworkers have developed a crack-resistant and tissue-mimetic hydrogel via a phase-transition-guided hierarchical engineering strategy that progressively constructs anisotropic polyvinyl alcohol networks. While previous studies have reported anisotropic PVA hydrogels, this work stands out by achieving a simultaneous and significant enhancement of multiple critical mechanical properties including strength, toughness, fatigue resistance, and notably, high stretchability (larger than 200% fracture strain with a tensile strength of ~60 MPa) by combining mechanical training, wet-annealing, and salting-out processes. I recommend this manuscript for publication after minor revisions, as it makes a meaningful and original contribution to the design of hierarchical hydrogels with improved mechanical and anti-swelling properties.

1: As known, the stretching method was often used to prepare anisotropic PVA hydrogels, and mechanical training of hydrogels in water was also reported to achieve the alignment of PVA macromolecules, and what are the unique advantages of the mechanical training of organogels and subsequently annealing in this study?

Response: The mechanical training of organogels combined with annealing in this study offers the following unique advantages compared to conventional stretching or mechanical training in water for PVA hydrogels: (1) More efficient regulation of molecular orientation: The mechanical training of organogels can be carried out rapidly in air, and only dozens of operations are needed to induce the directional arrangement of molecular chains to form an oriented structure. Higher stress transfer efficiency is achieved through the synergistic effect of solvent and polymer. In contrast, the mechanical training of traditional hydrogels is mostly performed underwater, with a more complex operation process, and the induction efficiency of molecular chain orientation is relatively low. (2) Better structural stability: The annealing strategy can further optimize the conformation of macromolecular chains. When wet annealing is applied to anisotropic organogels, it can promote the densification of the network structure, enhance intermolecular hydrogen bonding, and form more crystalline domains. Meanwhile, due to the presence of glycerol, a high-boiling-point solvent, it can ensure the thermal stability and structural stability of the organogel during the annealing process, which is difficult to achieve with traditional aqueous phase training methods. (3) Synergistically enhanced comprehensive performance: The combination of mechanical training and annealing of organogels realizes the synergistic effect of molecular orientation regulation and structural optimization. Compared with a single hydrogel stretching or aqueous phase mechanical training, it is more conducive to balancing the anisotropy and mechanical property stability of the material, providing a more efficient regulation path for the preparation of high-performance PVA-based functional materials. We have supplemented the aforementioned content in the Introduction section of the revised manuscript to more clearly highlight the innovative aspects of our methodology.

2: In many cases, achieving macromolecular alignment often compromises stretchability, and high strength and modulus typically result in low fracture strains (less than 100% for many strong anisotropic hydrogels). Interestingly, however, the PVA hydrogels in this work demonstrate a high strength of 61 ± 2.9 MPa while

maintaining a large fracture strain exceeding 200%. Please provide a discussion on the underlying mechanisms and the significance of this mechanical advantage.

Response: We agree with the reviewer that most anisotropic hydrogels exhibit a strain at break below 100%. However, the PVA hydrogel reported in this study achieves a much larger strain at break (>200%) while maintaining a high tensile strength of 61 ± 2.9 MPa. This unusual combination arises from the synergistic effects of mechanical training, wet annealing, and salt precipitation, as detailed below:

(1) Mechanical training of glycerol organogels: constructing an oriented molecular network. During mechanical training, PVA chains are progressively aligned along the applied force direction, forming a highly oriented molecular structure that provides the basis for high strength. The use of glycerol, a high-boiling-point solvent, ensures the stability of the organogel and allows slow and controlled chain alignment in a non-aqueous environment. This process preserves chain integrity and entanglements, thereby preventing the brittleness that often accompanies molecular orientation.

(2) Wet annealing: promoting crystallinity and chain entanglement. Wet annealing enhances intermolecular hydrogen bonding and facilitates the formation of additional crystalline domains, thereby reinforcing the network structure. Simultaneously, it promotes chain entanglement within the amorphous regions, which compensates for deformation capability and prevents premature fracture. Together, these effects improve network stability and reduce defects associated with molecular orientation, enabling the hydrogel to achieve higher strength while maintaining large fracture strain.

(3) Salt precipitation: locking orientation and regulating network density. Finally, the salting-out process strengthens hydrogen-bonding crosslinks and induces PVA chain self-aggregation through the Hofmeister effect, thereby further increasing crystallinity and network density. This step locks the anisotropic structure in place, preserves the orientation-derived reinforcement, and further enhances the load-bearing capacity of the hydrogel. Relevant discussion has been added into Section 2.2 of the revised manuscript.

3: The title emphasizes the role of progressive nanocrystallization; however, the crystallization process is primarily introduced during the wet-annealing step (as shown

in Fig. 1a). Please include a detailed description of how the crystalline structure evolves throughout the different stages of the fabrication process, and this will better elucidate the relationship between the processing stages and the development of the crystalline features that contribute to the hydrogel's enhanced mechanical properties.

Response: (1) During the mechanical training stage, the PVA chains align to form an anisotropic glycerol/PVA organogel (AV₁₅). The tensile strength of AV₁₅ reaches 45.0 MPa (Figure R1-1), showing a significant improvement compared to the isotropic organogel. The crystal size at this stage is approximately 4.48 nm (Figure R1-2).

(2) In the wet annealing stage, the conformation of the macromolecular chains is adjusted. The tensile strength of AV₁₅-120 increases to 68.92 MPa, higher than that of AV₁₅. The crystal size of AV₁₅-120 is about 6.2 nm. The diffraction peak of the (101) crystal plane in the XRD pattern (Figure R1-3) becomes significantly sharper, indicating an increase in crystallinity compared to AV₁₅. (3) During the salting-out stage, the molecular chains further aggregate, and the hydrogen bond network becomes denser. The tensile strength of AV₁₅S-120 reaches 79.17 MPa, showing an improvement over AV₁₅. The crystal size of AV₁₅S-120 is approximately 7.8 nm. The diffraction peak of the (101) crystal plane in the XRD pattern becomes sharper compared to that of AV₁₅-120, indicating a further increase in crystallinity. The dense hydrogen bonds and crystallization work synergistically to achieve energy dissipation. Through the progressive process from mechanical training to wet annealing and then to salting-out, the crystal size gradually increases, and the crystallinity also improves, leading to a stepwise enhancement of the mechanical properties.

Figure R1-1. Tensile stress-strain curves of AV₁₅, AV₁₅-120 and AV₁₅S-120.

Figure R1-2. Summary of average size of crystalline domains of AV₁₅, AV₁₅-120 and AV₁₅S-120.

Figure R1-3. X-ray diffraction (XRD) patterns of AV₁₅, AV₁₅-120 and AV₁₅S-120.

4: In Fig. 2, the annealing temperature significantly influences the final mechanical properties of the hydrogels. Notably, AV15SH-120 exhibited markedly higher tensile strength and fracture energy compared to AV15SH-90. Could you elaborate on how the optimal annealing temperature was determined? Specifically, what criteria or experimental considerations guided the selection of 120°C as the optimal temperature? Additionally, did the higher annealing temperature risk damaging or compromising the anisotropic structures within the hydrogels?

Response: We selected 120 °C as the optimal annealing temperature based on a comprehensive consideration of crystallinity, structural integrity, and mechanical performance. Annealing at elevated temperatures facilitates chain rearrangement, thereby enhancing crystallinity and chain entanglement. However, once the temperature exceeds a critical threshold, the excessive molecular mobility can disrupt or even destroy crystalline domains, resulting in a deterioration of mechanical properties. Moreover, overly high temperatures may compromise the anisotropic structure of the hydrogel. Consistent with these findings, a previous study on isotropic

PVA hydrogels also identified 120 °C as the optimal annealing temperature [Adv. Mater. 2023, 35, 2210624].

5: In Fig. 3b and 3c, an apparent orientation was observed in both AV15SH-120 and AV15H hydrogels. Please explain the underlying reasons for the significant difference in the degree of orientation between these two hydrogels?

Response: The significant difference in orientation degree between the two hydrogels originates from the regulation of molecular chain alignment and crystallite ordering by their distinct preparation routes. For AV₁₅SH-120, the high orientation results from the synergistic action of mechanical training, wet annealing, and salting-out. Mechanical training aligns PVA chains along the stretching direction to form an anisotropic structure. Wet annealing further densifies the network and strengthens intermolecular hydrogen bonding. Finally, salting-out “locks in” the oriented structure by promoting chain aggregation and increasing hydrogen bond density along the pre-oriented direction. These sequential steps collectively optimize chain alignment and crystallite ordering. In contrast, AV₁₅H undergoes only mechanical training. While stretching induces initial chain alignment, the absence of wet annealing and salting-out prevents further reinforcement of the oriented structure. Without ion-driven chain aggregation and enhanced intermolecular interactions, the oriented domains are difficult to maintain or improve upon immersion in water.

In summary, AV₁₅SH-120 achieves progressive optimization and stabilization of orientation through the sequence of “mechanical training → wet annealing → salting-out,” whereas AV₁₅H, relying solely on mechanical training, shows weaker intermolecular interactions and a lower degree of orientation.

6: In Fig. 4g, the crack did not propagate in the 5000th cycle, and why did it propagate along the fibril direction (crack deflection) rather than the notch direction? Please also explain why and how the stress was concentrated in the crystalline domains (Fig. 4i).

Response: The exceptional defect tolerance of the PVA hydrogel arises from its hierarchically orchestrated energy-dissipation architecture, which collectively

suppresses crack propagation even after long-term cyclic loading (e.g., 5000 cycles).

(1) Crack deflection mechanism.

When a crack initiates from the notch and is oriented perpendicular to the aligned fibril bundles, its initial propagation path crosses the fibers. However, the high-strength fibril bundles act as barriers that pin the crack tip. As a result, the crack preferentially deflects and propagates along the weaker inter-fibrillar interfaces rather than the original notch direction. This deflection significantly reduces the effective stress intensity at the crack tip. Moreover, the self-training effect near the crack tip allows localized molecular rearrangement, which relaxes stress concentration and further delays crack growth. These mechanisms explain why the crack does not propagate catastrophically even after 5000 cycles.

(2) Stress concentration in crystalline domains.

At the nanoscale, the fibril bundles contain directionally aligned nanofibrils and crystalline domains. The crystalline domains, owing to their high modulus and ordered molecular arrangement, serve as rigid crosslinking points. During stretching, stress is preferentially transferred from the softer amorphous matrix to these crystalline regions, leading to local stress concentration. Rather than being detrimental, this concentration enables the crystalline domains to function as effective energy-dissipating sites: they arrest or pin advancing cracks, enforce crack deflection, and stabilize the load transfer between nanofibrils. Simultaneously, nanofibrils bridge cracks and dissipate energy through pull-out and rupture, while dynamic hydrogen-bond breakage and reformation in the amorphous phase provide additional reversible energy dissipation.

In summary, the synergy of anisotropic fibril alignment, crystalline-domain reinforcement, and dynamic hydrogen-bond networks enables stress redistribution, crack deflection, and efficient energy dissipation. These multiscale mechanisms account for the hydrogel's exceptional resistance to crack growth under cyclic loading and explain why cracks deflect along fibril bundles instead of propagating directly from the notch. Relevant discussion has been added into Section 2.4 of the revised manuscript.

7: As shown in Fig. 5, the hydrogel exhibited excellent swelling resistance, and how did the anisotropic structure as well as the crystalline domains affect this performance? Additionally, do the anisotropic hydrogels exhibit advantages in swelling resistance compared to the isotropic counterparts?

Response: The excellent swelling resistance of the hydrogel (Figure 5) arises from the synergistic contributions of the anisotropic structure and crystalline domains, which together restrict water infiltration and network expansion. (1) Constraint from the anisotropic network. The uniaxially aligned PVA chains in the anisotropic hydrogel impose directional constraints on chain mobility and deformation, effectively suppressing volumetric expansion when exposed to water. This anisotropic restriction limits network relaxation and prevents excessive swelling. (2) Densification by crystalline domains. Crystalline domains, generated through sequential training, annealing, and salting-out, serve as rigid physical crosslinking points. Their dense packing, coupled with a high density of hydrogen bonds and reduced free volume, significantly lowers water penetration. The anisotropic arrangement of these crystalline domains further enhances resistance by reinforcing the network along the alignment direction.

Compared with isotropic counterparts, anisotropic hydrogels exhibit superior swelling resistance. Isotropic networks lack directional alignment, leading to irregular distribution of crystalline domains and the presence of “weak zones” that facilitate local water uptake and uneven swelling. In contrast, anisotropy ensures directional constraint and a more effective barrier to water infiltration.

2. Reviewer #2:

Recommendation: Reject

The manuscript proposes a “stepwise construction” strategy combining mechanical training, wet annealing, and salting-out was proposed to fabricate PVA hydrogels with hierarchical, anisotropic, and biomimetic structures. The reported mechanical properties (e.g., strength > 60 MPa, toughness > 100 MJ/m³, fracture energy > 85 kJ

m⁻² and fatigue threshold >2000 J/m²) are outstanding, representing a notable advance in tough hydrogel research. The authors attempt to elucidate structure–property relationships across multiple length scales (SEM, SAXS, WAXS, XRD and DSC) and present a schematic mechanism, reflecting a desire for in-depth understanding. However, the manuscript faces several critical issues: 1. Lack of rigor in data presentation and interpretation: Including inconsistencies between text and figures, incorrect legends, and imprecise terminology. 2. Insufficient support for key claims: The SAXS/WAXS/XRD analyses lack depth, and some critical intermediate states are not directly characterized, leaving the stepwise construction mechanism incomplete. 3. Conceptual novelty: While the performance is impressive, the methods themselves (mechanical training, annealing, salting-out) are not new. The manuscript should Clarify their unique synergy or irreplaceability in achieving this superior performance.

Response: Thank you for your comments.

1. We have carefully revised the main text, figures, and figure legends to ensure the accuracy and rigor of the data.
2. We have supplemented the characterization of key intermediate states (using SAXS/WAXS/XRD/SEM) to fully demonstrate the stepwise construction mechanism.
3. The combination of mechanical training of organogels, wet annealing, and salting-out in this study provides unique advantages over conventional stretching or mechanical training of PVA hydrogels in water, the novelty can be summarized as follows: (1) Efficient regulation of molecular orientation: Mechanical training of organogels can be performed rapidly in air, requiring only a few dozen cycles to induce directional alignment of polymer chains and form a highly oriented structure. The presence of the high-boiling-point solvent glycerol facilitates efficient stress transfer and chain mobility, significantly enhancing orientation compared to traditional hydrogel training in water, which is slower, more complex, and less effective at inducing molecular alignment. In addition, conventional dry-annealing of hydrogels (e.g., after water removal or aerogel formation) cannot sufficiently activate macromolecular mobility, which restricts the improvement of crystallinity and chain entanglement. In contrast, confined wet-annealing of organogels provides a more

favorable environment for chain rearrangement and conformational adjustment, as the presence of solvent reduces resistance to chain motion. This process thus promotes stronger hydrogen bonding and higher crystallinity. (2) Synergistic enhancement of comprehensive properties: The stepwise combination of mechanical training, wet annealing, and salting-out produces a progressive, cooperative reinforcement of the hydrogel network. Mechanical training aligns the chains, wet annealing optimizes crystallinity and chain entanglement in amorphous regions (compensating for deformation loss due to orientation), and salting-out further consolidates chain aggregation and network density, locking in the anisotropic structure. This indispensable multistep synergy, which has not been reported previously, enables the hydrogel to achieve outstanding mechanical performance together with excellent swelling resistance. We have incorporated this discussion into the Introduction of the revised manuscript to clearly emphasize the innovative aspects and cooperative significance of each processing step.

In conclusion, the manuscript is not yet suitable for publication in Nature Communications. There are some concerns (listed below) that should be addressed:

- 1.** The authors describe a “stepwise construction” process, but the structural changes after mechanical training, then after annealing, and the structural evolution from salting-out should be clearly illustrated. In Figure 1a, “Mechanical training” and “Annealing” are combined into a single arrow. If annealing (i.e., conformational optimization and crystallinity enhancement) is a critical intermediate step, separately depicting the state “Annealing” (i.e., following “Mechanical training”) would make the schematic more persuasive. Additionally, the transition from “anisotropic organogel” to “anisotropic hydrogel” in Figure 1a does not effectively depict the structural densification expected from the salting-out process.

Response: Thank you for your suggestion. We have revised the schematic diagram to more accurately illustrate the structural evolution process of mechanical training → annealing → salting-out → soaking in water. In the updated preparation diagram (Figure R2-1), the annealed state is depicted separately, and the structural densification is explicitly highlighted to differentiate the transition from the “anisotropic organogel”

to the “anisotropic hydrogel.” Furthermore, we have supplemented the manuscript with additional characterizations of the key intermediate states (SAXS/WAXS/XRD/SEM), which substantiate the stepwise construction mechanism and provide the basis for the modified preparation scheme. We have revised Figure 1 in the revised manuscript.

Figure R2-1. Schematic illustration of the fabrication of anisotropic hydrogels through mechanical training, wet annealing, and salt-locking.

2. The schematic clearly labels a “soaked in water” step and depicts distinct microstructural differences between the post-salting-out state (pink) and the final hydrogel (green). However, this structural transition is not discussed in the text. This omission could lead readers to question the “structure locking” mechanism: specifically, at which step does structural densification actually occur? If densification happens during this “soaked in water” phase, rather than during “salting-out,” it would contradict the authors proposed core mechanism of “structure locking” by Hofmeister effect-driven salting-out.

Response: Thank you for pointing out this issue. We have revised the text to clearly describe this transition process and confirm its consistency with the core mechanism, as follows:

The “soaking in water” step in Figure 1a is designed to remove residual salts from the hydrogel network after salting-out, as mobile ions can readily migrate out and severely restrict biomedical applications by causing cell dehydration and rupture. To overcome

this issue, the material was converted into a PVA hydrogel, thereby broadening its application potential. It should be emphasized that structural densification occurs during the salting-out step rather than in the subsequent water-soaking process. To substantiate this, we have supplemented XRD and WAXS data for the mechanically trained, annealed, and salted-out hydrogel (AV₁₅S-120). As shown in the XRD patterns (Figure R2-2), AV₁₅S-120 exhibits a stronger and sharper crystalline peak at $2\theta = 20.2^\circ$ compared to AV₁₅SH-120, indicating increased crystallinity induced by the salting-out process, which diminishes after water soaking. These results are consistent with the observed mechanical properties. Furthermore, the azimuthally integrated intensity distributions confirm the oriented structures of the hydrogels (Figure R2-3). The azimuthal intensity profiles of AV₁₅S-120 and AV₁₅SH-120 are nearly identical, and the calculated orientation degrees (0.90 and 0.87, respectively) demonstrate that the orientation established during the salting-out step is largely preserved after water soaking. This process is consistent with our core mechanism: the Hofmeister effect during salting-out induces molecular chain aggregation and microcrystal growth, locks in the orientation, and forms a dense oriented structure. We have added relevant explanations in Section 2.1 of the manuscript, emphasizing that densification is attributed to the salting-out step rather than the "soaked in water" step.

Figure R2-2. X-ray diffraction (XRD) patterns of AV₁₅S-120 and AV₁₅SH-120.

Figure R2-3. Azimuthally integrated intensity distribution of 2D WAXS patterns of AV₁₅S-120 and AV₁₅SH-120.

3. It is noticed that the water content changes significantly after the “stepwise construction”. The mechanical properties of hydrogels are closely related to their water content, as a decrease in water content (i.e., an increase in polymer concentration) inherently leads to increased strength and modulus. To demonstrate that the observed enhancements arise from the proposed structural optimization rather than simply from concentration effects, it is recommend providing mechanical property data normalized by polymer content, along with a comparative analysis and discussion of the results before and after normalization.

Responses: To clarify that the observed performance enhancements result from structural optimization rather than solely from concentration effects, we supplemented the analysis with normalized mechanical properties by polymer content. We normalized the raw data by dividing by the polymer content, representing the mechanical properties per 1% polymer concentration. Figure R2-4 presents the unnormalized strength and modulus data, showing that both stepwise treatment and increased concentration contribute to improved mechanical performance, thereby necessitating a distinction between these two factors. The normalized results (Figure R2-5) demonstrate that, even after excluding the influence of polymer concentration, the strength and toughness of AV₂₀SH-120 are still 46-fold and 222-fold higher than those of IV₁₀SH, respectively.

These findings confirm that the substantial performance enhancement originates primarily from the proposed structural optimization strategy. Relevant discussion has been incorporated into Section 2.2 of the revised manuscript.

Figure R2-4. Summary of strength and modulus of different hydrogels.

Figure R2-5. Summary of strength and modulus of different hydrogels normalized by polymer content.

4. In Figure 3c, the 2D SAXS and WAXS patterns of the AV₁₅SH-120 sample show clearly different scattering orientations: the SAXS signal is stronger in the vertical direction, while the WAXS signal is more concentrated horizontally. This appears

inconsistent with Figures 3d and 3f. Could the authors clarify whether this arises from differences in data processing or from orientation differences at different structural length scales? Additionally, it is suggest indicating the stretching direction in Figure 3c.

Response: Thank you for highlighting this issue. The discrepancy arose from differences in sample orientation during test preparation. We have reprocessed the WAXS data to ensure consistency with the SAXS results, and the stretching direction has now been clearly indicated in Figure 3c of the revised manuscript.

5. In Figure 3e, for AV15SH-120, there's another smaller peak around $q = 1.625 \text{ \AA}^{-1}$. It is suggest discussing whether this peak may originate from another crystallographic plane or a higher order reflection, as this would help readers better understand the evolution of the crystalline structure. Furthermore, the q -values of IV15SH and AV15H appear very close in the figure. Could the authors clarify the method used (e.g., peak fitting) to accurately distinguish the q_{max} of these two peaks? Providing the fitting results or explain more about how you found these peak positions would strengthen the reliability of the conclusions.

Response: (1) After the annealing step, the hydrogel exhibited a small diffraction peak at $q \approx 1.625 \text{ \AA}^{-1}$ ($2\theta \approx 23^\circ$), corresponding to the (200) crystal plane of the PVA crystalline structure [Zhang, J., Zhang, M., Wan, H. et al. Coordinatively stiffen and toughen polymeric gels via the synergy of crystal-domain cross-linking and chelation cross-linking. *Nat Commun.* 16, 320 (2025)], which was nearly undetectable in the non-annealed hydrogel. The emergence of this peak can be attributed to the enhanced chain mobility during wet annealing, which facilitates conformational rearrangement of macromolecules, elimination of structural defects, and promotion of crystallite growth. The salting-out step further strengthens intermolecular hydrogen bonding and promotes tighter chain packing, leading to an increase in crystallinity and higher intensity of this (200) reflection. Therefore, the appearance and enhancement of this small diffraction peak can be ascribed to the synergistic effects of wet annealing and salting-out. A similar phenomenon was also observed in (Figure R2-6), confirming this interpretation. Relevant discussion has been incorporated into Section 2.3 of the revised manuscript.

(2) Regarding the proximity of the q_{\max} values for AV₁₅SH and AV₁₅H, we acknowledge that their positions are indeed very close in the WAXS profiles (Figure R2-7) and can be considered essentially unchanged. For data processing, the WAXS patterns were exported using Diffrac.SAXS software and subsequently analyzed in Origin, where the built-in integration and analysis functions were employed to determine the q_{\max} values directly, rather than through peak-fitting procedures. We have clarified this point in the revised supporting information to strengthen the reliability of the conclusions.

Figure R2-6. 1D WAXS curves showing scattering intensity vs. scattering vector (q).

Figure R2-7. 1D WAXS curves showing scattering intensity vs. scattering vector (q).

6. Figure 3g presents the Lorentz-corrected (Iq^2) 1D SAXS profiles, which is effective for analyzing long-period structures (L) and the evolution of lamellar features. However,

However, to offer readers a more comprehensive understanding of the sample's original scattering information and potentially reveal other structural features, it is recommended authors providing the raw, uncorrected 1D SAXS scattering curves, either in the main text or the Supporting Information. Such an addition typically helps readers better grasp the overall characteristics of the scatterers and the original state of the data before processing.

Response: We have added the raw, uncorrected 1D SAXS scattering curves (Figure R2-8) in the Supporting Information, which correspond to the Lorentz-corrected (Iq^2) 1D SAXS profiles in Figure 3g. The raw curves clearly show the scattering intensity distribution of different samples in the low to high q range, and can more directly reflect the original variation trend of the scattering signals. These revisions have been integrated into the manuscript.

Figure R2-8. 1D SAXS curves, depicting scattering intensity vs. scattering vector (q).

7. It is recommended that the authors clearly indicate whether the 1D SAXS/WAXS profiles are obtained from integration parallel or perpendicular to the nanofiber alignment direction, as the anisotropic structure would yield significantly different scattering features in each orientation. It is further recommended that the authors include specific details on the integration method and direction used for this data in all

relevant 1D SAXS/WAXS figure captions and corresponding descriptions in the main text.

Response: The 1D WAXS profiles are obtained from integration parallel to the nanofiber alignment direction. We have clearly indicated the integration direction in all relevant figure captions of 1D SAXS/WAXS and the descriptions in revised manuscript.

8. The authors interpret “L” as the average distance between adjacent crystalline domains, stating that the decrease in L, defined as the decrease in average distance between adjacent crystalline domains, as evidence of polymer chain densification. While this interpretation is plausible, L can also be influenced by changes in crystal morphology, grain size, or the number of crystalline nuclei. Considering this complexity, attributing the decrease in L directly and solely to “densification between polymer chains” may lack sufficient rigor. It is suggested the authors consider incorporating WAXS data to provide more specific and direct evidence for this claim of “densification between polymer chains”.

Response: Thank you for this insightful suggestion. We fully agree that the parameter *L* can be influenced by multiple factors, including crystal morphology, grain size, and the number of crystalline nuclei. In the revised manuscript, we supplemented the data, and we have clarified that polymer chain densification results from the synergistic effects of macromolecular chain alignment and crystalline evolution, rather than being solely attributed to the decrease in *L*.

SEM images confirm the formation of densified fibrillar structures. WAXS results demonstrate a gradual increase in orientation degree during the stepwise fabrication process. To comprehensively verify chain densification from the crystallization perspective, we have incorporated combined results from DSC, WAXS, SAXS, and XRD analyses (Table R1). DSC results reveal a gradual increase in crystallinity. XRD results show a progressive growth in crystal size across the anisotropic hydrogels. SAXS analysis indicates that the parallel inter-domain distance decreases significantly from 17.4 nm to 10.3 nm, whereas the perpendicular inter-domain distance decreases significantly from 18.4 nm to 8.5 nm, clearly demonstrating salting-out-induced chain aggregation and densification. Meanwhile, WAXS data show that the average chain

spacing and interlayer distance remain essentially unchanged, thereby excluding their contribution.

Taken together, these multi-technique results confirm that polymer chain densification primarily arises from macromolecular chain alignment, increased crystallinity and crystal size, and reduced inter-domain spacing. These clarifications have been incorporated into Section 2.3 of the revised manuscript.

Sample	Crystal dimension (D, nm)	Inter-crystal spacing (L_1 , nm)	Inter-crystal spacing (L_2 , nm)	Inter-laminar spacing (L_3 , nm)	Inter-molecular spacing (L_4 , nm)	Crystallinity (%)	Orientation degree
IV ₁₅ SH	5.9	18.4	17.4	0.42	-	12.63	0
AV ₁₅ H	6.1	16.1	15.2	0.43	0.7	14	0.5
AV ₁₅ SH-120	7.6	8.5	10.3	0.46	0.7	34.3	0.87

Table R1. Summary of crystal dimension (D), inter-crystal spacing (L_1), inter-crystal spacing (L_2), inter-laminar spacing (L_3), inter-molecular spacing (L_4), crystallinity and orientation degree within the PVA hydrogel samples.

9. In the XRD patterns shown in Figure 3i, AV15SH-120 exhibits not only a sharp and intense main peak at $2\theta \approx 19.7^\circ$, but also a secondary peak at higher diffraction angles (e.g., around $2\theta \approx 22-23^\circ$), similar to what is observed in the WAXS data. This observation prompts the question of whether new, or significantly enhanced, crystalline plane has formed in the PVA gel after the series of treatments involving mechanical training, high-temperature annealing, and salting-out. It is suggested that the authors provide a more detailed discussion and assignment of this secondary peak in the manuscript. Furthermore, the possible relationship between the emergence of this new crystalline plane and the observed exceptional mechanical properties warrants further exploration.

Response: After the annealing step, the hydrogel exhibited a weak diffraction peak at $2\theta \approx 23^\circ$ (Figure R2-9), corresponding to the (200) crystal plane of the PVA crystalline structure, which was nearly undetectable in the unannealed hydrogel. This peak emergence is attributed to enhanced molecular chain mobility during wet annealing, which facilitates optimized chain conformations and eliminates internal defects. Notably, the peak intensity further increases after the salting-out treatment, likely due to closer molecular packing and strengthened hydrogen bonding interactions induced by the salting-out process. Thus, the appearance of the (200) diffraction peak can be ascribed to the synergistic effects of annealing and salting-out. The development of this new crystalline plane contributes to the increased crystallinity and, consequently, the improved mechanical properties. Furthermore, comparison among hydrogels annealed at different temperatures reveals that higher annealing temperatures enhance the (200) peak intensity, consistent with the observed improvements in mechanical performance.

Figure R2-9. X-ray diffraction (XRD) patterns of different hydrogels.

10. The manuscript refers to the formation of “proto-crystalline regions” and state that the final material possesses “crystalline domains”. It is would appreciate clarification on how do the authors define “proto-crystalline regions”? What are the specific structural distinctions between these and the more stable “crystalline domains” ultimately observed? Furthermore, can the existing characterization data (e.g., XRD/WAXS) differentiate or provide evidence for the existence and evolution of these

distinct crystalline states? It is observed that neither the schematic for the fabrication process (Figure 1a) nor that for structural characterization (Figure 3) clearly depicts “proto-crystalline regions” or their transformation into the final “crystalline domains”. Therefore, it is suggest considering revisions to these schematics (Figure 1a) to more explicitly illustrate the evolution of the crystalline structure.

Response: We define the proto-crystalline domains as the crystalline regions within the isotropic organogel formed after solvent exchange. At this initial stage, the organogel (IV₁₅) exhibits a primitive crystallization morphology with relatively low crystallinity. During the subsequent stepwise fabrication, the crystal size and overall crystallinity progressively increased, while the distance between adjacent crystalline regions decreased (see detailed information in Table R1 and following section). We have revised the schematic illustration in Figure 1. For clarity, the term “proto-crystalline domains” has been replaced with “primitive crystalline domains.”

11. The manuscript emphasizes a “stepwise construction” strategy involving mechanical training, annealing, and salting-out. While the authors compare the structures of different final states (e.g., IV₁₅SH, AV₁₅H, AV₁₅SH-120), it is strongly recommended that they also include structural characterization of intermediate structures after each key processing step: (1) post mechanical training only; (2) post-training and annealing; and (3) post-training, annealing, and salting-out. To achieve this, it would be beneficial to provide characterization data (such as SEM, WAXS, SAXS, XRD, and DSC). This approach would more clearly establish the direct link between each processing stage and the resulting structural changes, thus offering more robust support for “stepwise construction” mechanism.

Response: In the revised manuscript, we have supplemented the characterization of intermediate structures and mechanical properties after each key processing step as follows: (1) post mechanical training only (AV₁₅); (2) post-training and annealing (AV₁₅-120); and (3) post-training, annealing, and salting-out (AV₁₅S-120).

(1) We have supplemented the XRD tests (Figure R2-10). All PVA gels exhibit a diffraction peak near $2\theta = 19.7^\circ$. The peak of AV₁₅ is small and broad with low intensity

(Figure R2-10a), whereas AV₁₅S-120 displays a distinct crystalline peak, indicating an increase in the crystallinity of the PVA gel. Furthermore, we calculated the average crystal size (D), which increased from 4.48 nm to 7.8 nm (Figure R2-10b).

Figure R2-10. (a) X-ray diffraction (XRD) patterns of AV₁₅, AV₁₅-120 and AV₁₅S-120. (b) Summary of average size of crystalline domains of AV₁₅, AV₁₅-120 and AV₁₅S-120.

(2) We have supplemented the WAXS tests (Figure R2-11). The 2D WAXS patterns consistently exhibited anisotropy with gradually intensifying diffraction arcs (Figure R2-11a), confirming the formation and progressive enhancement of anisotropic structures during the stepwise fabrication process. The azimuthal intensity distribution profile of AV₁₅S-120 is noticeably sharper than that of AV₁₅, reflecting a higher degree of orientation induced by salting-out, which effectively locks the molecular alignment (Figure R2-11b). The high orientation degree in the anisotropic gels is primarily derived from the mechanically induced alignment of PVA chains, while the salting-out step further stabilizes this oriented architecture. In addition, the strongest peak in the WAXS 1D profile appears at $q = 1.57 \text{ \AA}^{-1}$ (Figure R2-11c), corresponding to the average interchain spacing between adjacent PVA chains in nanocrystalline regions, while the weaker peak at $q = 0.89 \text{ \AA}^{-1}$ represents the average interlayer spacing within nanocrystalline domains; notably, the stepwise preparation exerts little influence on these two crystalline parameters.

Figure R2-11. (a) Wide-angle X-ray scattering (WAXS) patterns of AV₁₅, AV₁₅-120 and AV₁₅S-120. (b) Azimuthally integrated intensity distribution of 2D WAXS patterns. (c) 1D WAXS curves showing scattering intensity vs. scattering vector (q).

(3) We have supplemented the SAXS tests (Figure R2-12). As shown in Figure R2-12a, the 2D SAXS patterns of AV₁₅, AV₁₅-120 and AV₁₅S-120 exhibit clear anisotropy. In the parallel direction (chain alignment), the average inter-domain distance varies only slightly. In the perpendicular direction, the average inter-domain distance decreased from 9.39 nm to 8.36 nm and further to 6.30 nm (Figure R2-12f), indicating that salting-out promotes structural densification of the gel. We summarized crystal dimension (D), inter-crystal spacing (L_1), inter-crystal spacing (L_2), inter-laminar spacing (L_3), inter-molecular spacing (L_4) and orientation degree of AV₁₅, AV₁₅-120 and AV₁₅S-120 in Table R2.

Figure R2-12. (a) Small-angle X-ray scattering (SAXS). (b) 1D SAXS curves (Perpendicular to the orientation direction, V_{\perp}), depicting scattering intensity vs. scattering vector (q). (c) 1D SAXS curves (V_{\perp}), depicting scattering intensity vs. scattering vector (Iq^2). (d) 1D SAXS curves (Parallel to the orientation direction, $V//$), depicting scattering intensity vs. scattering vector (q). (e) 1D SAXS curves ($V//$), depicting scattering intensity vs. scattering vector (Iq^2). (f) Calculated average distance between crystalline regions.

Sample	Crystal dimension (D, nm)	Inter-crystal spacing (L_1 , nm)	Inter-crystal spacing (L_2 , nm)	Inter-laminar spacing (L_3 , nm)	Inter-molecular spacing (L_4 , nm)	Orientation degree
AV ₁₅	4.48	9.39	10.06	0.4	0.7	0.82
AV ₁₅ -120	6.2	8.36	11.43	0.4	0.7	0.86
AV ₁₅ S-120	7.8	6.30	10.30	0.4	0.7	0.90

Table R2. Summary of crystal dimension (D), inter-crystal spacing (L_1), inter-crystal spacing (L_2), inter-laminar spacing (L_3), inter-molecular spacing (L_4) and Orientation degree.

(4) We have supplemented the SEM tests (Figure R2-13). Since the organogel cannot be directly characterized by conventional SEM imaging, we used SEM images of the corresponding final-state hydrogels to represent the oriented structures of the intermediate states. The manuscript already includes SEM characterizations of hydrogels subjected to mechanical training only and to the combined mechanical training - annealing - salting-out process. We have also updated the SEM images in Section 2.3. Specifically, AV₁₅H exhibits an oriented fibrous structure along the training direction (Figure R2-13a), whereas AV₁₅H-120 displays a denser and more highly aligned fibrous architecture (Figure R2-13b), and the highly orientated structure is well maintained for AV₁₅SH-120 (Figure R2-13c). These results confirm that mechanical training initiates structural orientation in the hydrogel, wet annealing promotes tighter packing of the oriented structure, and salting-out treatment further locks in this molecular alignment.

Figure R2-13. SEM images of (a) AV₁₅H, (b) AV₁₅H-120 and (c) AV₁₅SH-120.

(5) We have supplemented the transmittance characterization (Figure R2-14). We have also supplemented UV transmittance measurements of the intermediate gels. The transparency of AV₁₅-120 is slightly lower than that of AV₁₅, which can be attributed to the crystallinity increase induced by annealing, whereas AV₁₅S-120 exhibits a much larger decrease in transparency. This pronounced reduction is ascribed to salting-out-induced chain aggregation, which can be regarded as a form of partial phase separation.

Figure R2-14. UV-vis transmittance spectra showing the transparency of AV₁₅, AV₁₅-120 and AV₁₅S-120.

(6) We have supplemented the mechanical property tests (Figure R2-15). In this work, the stepwise preparation strategy was designed to regulate orientation and crystallization, thereby enhancing the mechanical performance of the final hydrogels. Figure R2-15 presents the mechanical properties of the gels at different intermediate states. After mechanical training, AV₁₅ exhibited a tensile strength of 44.33 ± 1.34 MPa, elongation at break of $219.85 \pm 27.61\%$, elastic modulus of 43.26 ± 3.69 MPa, and toughness of 62.51 ± 10.64 MJ/m³. Following annealing, AV₁₅-120 demonstrated improved properties, with a tensile strength of 69.28 ± 1.94 MPa, elongation at break of $268.67 \pm 48.22\%$, elastic modulus of 61.28 ± 10.17 MPa, and toughness of 130.85 ± 19.34 MJ/m³. After salting-out, AV₁₅S-120 further improved to a tensile strength of 77.44 ± 2.07 MPa, elongation at break of $296.13 \pm$

17.04%, elastic modulus of 59.78 ± 3.66 MPa, and toughness of 156.34 ± 11.47 MJ/m³. These results clearly demonstrate that the mechanical properties are progressively enhanced through the sequential steps of mechanical training, annealing, and salting-out.

Figure R2-15. (a) Tensile stress-strain curves of AV₁₅, AV₁₅-120 and AV₁₅S-120, with a summary of their (b) tensile strength and fracture strain, and (c) modulus and toughness.

In summary, the combined XRD, WAXS, SAXS, SEM, UV transmittance, and mechanical testing results consistently demonstrate that each processing step contributes uniquely: mechanical training induces initial orientation, annealing enhances crystallinity and densifies the network, and salting-out locks the oriented structure through chain aggregation and strengthened interactions. These findings provide robust experimental evidence supporting the effectiveness of the proposed stepwise construction strategy.

12. While the fatigue threshold (Γ_0) is a standard concept, the authors do not explicitly state in the manuscript how they specifically defined or extracted this value from the dc/dN vs. G data in Figures 4a-e. For instance, it is unclear whether Γ_0 was determined by extrapolation to $dc/dN=0$. Furthermore, the number of data points in the plots appears limited, the linear fits are not convincing, and some exhibit poor linearity, and the overall fitting accuracy seems questionable.

Response: A single-notch method to determine the fatigue resistance of the hydrogels. To prevent dehydration of the hydrogel, all fatigue tests were conducted in a water bath with the initial crack length (c_0) of the hydrogel, less than one-fifth of the specimen

width (L_0). The hydrogels were performed using a customized mechanical stretcher (FULETEST, China) equipped with a 100 N loading cell. A digital camera (AF4915ZTL, Dino-Lite) was used to record the initial crack propagation over continuous stretching cycles without a relaxation. The curves of nominal stress S versus stretch λ of the unnotched samples were obtained over N th cycles with the maximum applied stretch of λ_{max} . The strain energy density of the unnotched sample under the N th cycle with the maximum applied stretch of λ_{max} can be calculated as: $W(\lambda_{max}, N) = \int_1^{\lambda_{max}} S d\lambda$. The applied energy release rate G in the notched sample under the N th cycle with the maximum applied stretch of λ_{max} can be calculated as: $G(\lambda_{max}, N) = 2k(\lambda_{max}) \cdot c(N) \cdot W(\lambda_{max}, N)$. where k is a slowly varying function of the applied stretch as $k = 3/\sqrt{\lambda_{max}}$. c and W refer to the crack length and the integral area of the N th loading part. In our analysis, the fatigue threshold (Γ_0) was obtained by linearly extrapolating the dc/dN - G curve to $dc/dN = 0$, which is a standard approach reported in the literature. We agree that the determination of Γ_0 is sensitive to testing conditions. To minimize dehydration of the hydrogels, all fatigue tests were performed under water. However, long term immersion inevitably led to partial or slight swelling, which alters the mechanical response. Moreover, cyclic stretching during testing may further influence the swelling behavior and, consequently, the data scatter. Similar phenomena have been documented in others' studies [G. Su, X. Zhang, Y. Zhou, et al. "Biomimetic All-Weather Strong, Tough, and Fatigue-Resistant Composite Organohydrogels for Electronic Artificial Ligaments." *Small* (2025): e04139.]. Despite these limitations, the extracted values exhibit consistent trends across different samples, which supports the reliability and reasonableness of our results. In our future work, we will seek to modify the fatigue testing setup to minimize swelling effects and achieve more accurate measurements.

13. For all relevant mechanical property test results (e.g., tensile strength, fracture energy, fatigue tests, etc.), it is also recommended that the authors specify whether the tests were conducted parallel or perpendicular to the nanofiber orientation.

Response: Thank you for your suggestion. All mechanical tests were performed parallel to the nanofiber alignment, and we have revised the Experimental section to clearly indicate this.

The manuscript still contains numerous instances of imprecise or nonstandard expressions:

1. Line 119, in the sentence “Hofmeister-specific ion coordination during salting-out,” the use of the term “coordination” is inappropriate. The Hofmeister effect primarily refers to how specific salt ions influence polymer solubility, aggregation behavior, and phase states by modulating solvent structure (especially water) and non-covalent interactions between solutes (such as electrostatic, hydrophobic, and hydrogen bonding interactions). This process typically does not involve the formation of coordination bonds in a chemical sense.

Response: Thank you very much for your suggestion. We have revised the original text accordingly. The sentence “Hofmeister-specific ion coordination during salting-out” has been replaced with: “Hofmeister-specific ion effects interactions during salting-out”.

2. Line 174, The statement “Wet annealing introduces temperature-dependent conformational optimization” is not sufficiently rigorous. Based on the overall context of the manuscript, wet annealing not only facilitates conformational rearrangements but also leads to structural changes.

Response: In the revised manuscript, we have changed the statement “Wet annealing introduces temperature-dependent conformational optimization” to “Wet annealing optimizes macromolecular conformations, eliminates internal defects, enhances intermolecular hydrogen bonding, promotes the formation of more crystalline domains, and increases the degree of crystallinity.”

3. In Figure 2a, the sample name is incorrect: “AV15H-120” should be “AV15SH-120”.

Response: We have revised Figure 2a in the revised manuscript.

4. In Figure 3e, the authors state that, “IV15SH、 AV15H and AV15SH-120 appear at 0.46 nm, 0.43 nm, and 0.42 nm, respectively”. However, in the figure, the authors have labeled the strongest peak indicated was 0.46 nm for AV15SH-120.

Response: We sincerely apologize for this oversight. IV₁₅SH, AV₁₅H and AV₁₅SH-120 appear at 0.42 nm, 0.43 nm, and 0.46 nm, respectively. We have revised it in the revised manuscript.

3. Reviewer #3 (Remarks to the Author):

In their submitted Manuscript, the authors present very valuable results, which on the Reviewer's opinion are of considerable interest for the readers of Nature Communications. The work surely will achieve a prominent status in the fields of biomimetic materials, of advanced anisotropic hydrogels, and of materials design.

In the present study, the authors obtained bio-mimetic (tendon-like) hydrogels which display exceptionally good mechanical properties in their equilibrium-swollen state (and are resistant to any further swelling). In their gel preparation design, the authors were able to overcome the trade-off strength vs. toughness, and also swelling vs. mechanical properties, which is a great success.

The obtained hydrogels markedly surpass previously developed anisotropic gels of similar type, described in literature, and they even surpass advanced biological tissues in some critical properties.

A notable success consists in the achieved excellent fatigue resistance of the studied hydrogels, which is related to intrinsic mechanisms of crack pinning and deflection, as well as to energy dissipation via disruption/recombination of sacrificial hydrogen bonds. The characteristics of the gels (including biocompatibility tests) suggest a great potential in biomedical applications, but possibly also in technical ones, such as soft robotics.

Of considerable own scientific value is the preparation procedure itself, which in a hierarchical manner builds-up a tendon-like anisotropic structure, and thus the mechanical strength and toughness of the hydrogels, in the sequence: Crosslinking of polyvinyl alcohol by initial partial crystallization via freeze drying, mechanical training by cyclic stretching in the presence of a poor plasticizer solvent, further crystallization and re-organization via annealing in the same solvent, solvent exchange for water, followed finally by an enhancement of hydrogen bonding via a salting-out treatment.

The obtained products display unusually attractive properties, and were well-characterized. The processing strategy which yielded the gels is very sound, and is of great scientific value for developing new advanced bio-mimetic materials in the future. The paper is well-written, meets high scientific standards, and all its conclusions are well-supported by data. The Experimental Section also is well-written and provides sufficient information for work reproduction in general, albeit some improvements are suggested below.

For all the above reasons, the Reviewer highly recommends the Manuscript for publication in Nature Communications.

Below are listed some minor technical issues, which the Reviewer recommends to be addressed (this can be done quickly), in order achieve the maximum impact on the reader, and optimal information:

1) Experimental Section, lines 504–508:

The current description of gel annealing appears confusing, also in light of later discussion of gel synthesis and characterization in the Manuscript: After obtaining AV15SH (line 504) was the material really annealed as aqueous gel, immersed again in sodium citrate and finally soaked again in water (as it seems to be stated)? All the later discussion suggests that the authors inserted annealing as a new process step (after training, but before citrate treatment), in order to obtain the product series AV15SH-x. The most reader-friendly solution would be to introduce a new paragraph “(3) Preparation of annealed anisotropic hydrogels” for describing the preparation of AV15SH-x.

Response: Thanks for the valuable suggestion. We have added two new paragraphs in the revised manuscript to describe the preparation of anisotropic hydrogels as follows: **(3) Preparation of trained and annealed anisotropic gels:** The organogels prepared as described in underwent mechanical training for 30 cycles at a strain of 300% and a tensile rate of 200 mm/min. Then, wet annealing was performed at 120 °C for 30 min to obtain AV_x-y, where x indicates the initial PVA mass fraction, and y denotes the wet-annealing temperature. The gel (AV_x-y) was then directly soaked in deionized water for 48 h to obtain AV_xH-y.

(4) Preparation of trained, annealed and salting out anisotropic gels: The organogels prepared as described in underwent mechanical training for 30 cycles at a strain of 300% and a tensile rate of 200 mm/min. Then, wet annealing was performed at 120 °C for 30 min. Subsequently, it was immersed in a sodium citrate solution for 12 h to obtain AV_xS-y, followed by soaking in water for another 48 h to obtain AV_xSH-y, where x denotes the initial PVA mass fraction, and y denotes the wet-annealing temperature.

2) Introduction:

line 39: “Additionally, hydrogels exhibit excellent biocompatibility, ...”

and line 42: “However, their inherently nonuniform polymeric network ...”

suggested would be more precise comments, e.g.

line 39  “Additionally, SOME hydrogels ... etc. ...”

and line 42  “However, their OFTEN inherently ... etc. ...”

Response: Thank you for your insightful comments. we have revised it to:

"Additionally, some hydrogels exhibit excellent biocompatibility, ..." and "However, their often inherently nonuniform polymeric network ...". We have revised it to the revised manuscript.

3) 2.1. Design of anisotropic hydrogels: line 100, Figure 1:

Figure 1c: suggested is some more distinct separation of the data referring to Modulus and to Stress, maybe a vertical line dividing the graph;

Figure 1d is a very intuitive one, but a version of better quality (resolution) or with higher contrast would be useful for the final version of the paper.

Response: We have revised the figures according to the comments. Specifically, in Figure 2c, a vertical dividing line has been added to clearly separate the data regions for Modulus and Stress, thereby making the comparison between the two datasets more intuitive. In addition, Figure 2d has been replaced with a higher-resolution version to ensure that all details are clearly visible. These modifications improve the readability and clarity of the figures, in line with your suggestions.

4) (2.1. Design of anisotropic hydrogels): adding a few words would be useful at the following lines:

line 114/115:

“Subsequent mechanical training under controlled ethanol volatilization ...”

a more specific description would be of interest for the reader at this place:  e.g.

“Subsequent mechanical training (30 cycles of stretching and relaxation up to 300%) under ... etc. ...”;

how was the volatilization controlled?

was there some significant creep during the training?

line 117:

“Wet annealing enables ...”  e.g. “Wet annealing IN GLYCEROL AT 90 OR AT 120°C enables ...”

line 119/120:

“... during salting-out, where ...”  e.g. “... during salting-out WITH SODIUM CITRATE, where ...”

Response: We have added “Subsequent mechanical training (30 cycles at a strain of 300% and a tensile rate of 200 mm/min) under ambient-condition” to the section 2.1 of revised manuscript.

Since the mechanical training was conducted in air, the ethanol had already volatilized during the training process. And there was no some significant creep during the training. We have revised it to "Wet annealing in glycerol at 90°C or 120°C for 30 min enables." and " ... during salting-out with 1.5 M sodium citrate solution, where ... "

5) (2.1. Design of anisotropic hydrogels):

line 128 (appearance of the final product): “... yielding transparent hydrogels (Figure S1, ...”:

a photograph of the gel would be desirable rather in Figure 1 (line 100) than in Figure S1; with optimal illumination, very impressive images of highly transparent gels (like the studied ones) can be obtained.

Response: Thank you very much for your suggestion. However, after comprehensively considering the core information of Figure 1, we prefer to retain the photograph of the transparent hydrogel in Supplementary Figure S1 for the following reasons: Figure 1 is a "schematic diagram of the stepwise construction process," whose core purpose is to illustrate the logical structural evolution across key steps (such as mechanical training, annealing, and salting-out), including schematic representations of microstructural changes and transformation relationships between steps. Supplementary Figure S1, serving as a characterization figure for transparency, is more appropriate for collectively presenting physical photographs and their corresponding transparency data.

6) (2.1. Design of anisotropic hydrogels):

“(orientation factor >0.6) for ...”

suggested:  “(orientation factor >0.6 , as observed by WAXS, SAXS) for ...”

Response: We have recalculated the degree of orientation and revised it to "(orientation factor >0.8 , as observed by WAXS) for ...".

7) 2.2. Mechanical properties: discussion text (lines 151-225):

It would be useful to describe the family of the studied products, including the meaning of codes like “IV15SH”, etc., either at the begin of this chapter, or at the end of the previous one (2.1. Design of anisotropic hydrogels).

Response: For clarity in descriptions and discussions, the following designations are used: IVxSH refers to isotropic hydrogels prepared via wet-annealing and salting out; AVxH denotes anisotropic hydrogels obtained through mechanical training; AVxSH represents anisotropic hydrogels prepared by combining mechanical training and salting out; AVxSH-y specifies anisotropic hydrogels fabricated through mechanical training, wet-annealing, and salting out, where x indicates the initial PVA mass fraction, and y denotes the wet-annealing temperature. We have described these implications in Section 2.2 of the manuscript.

8) 2.2. Mechanical properties: discussion text (lines 151-225):

The reviewer would recommend to split the discussion into more paragraphs, and to arrange them in a sequence following the steps of the gels preparation (and their

mechanical/structure consequences), plus commenting additional aspects like: PVA concentration, Resistance to crack propagation, Comparison with biological tissues, Comparison with competing mechanically trained systems.

In the present text, new paragraph breaks would be suggested at lines 165, 167, 191, 203 and 212, in order to highlight their topics (ideally, the first words in each paragraph should give a quick orientation).

Response: We have split the discussion into more paragraphs. The discussions regarding PVA concentration, comparison with biological tissues, and comparison with competing mechanically trained systems have been highlighted in red in the revised manuscript. Further discussion on crack propagation resistance is presented in section 2.4 of the manuscript.

9) 2.3. Characterization of the microstructures:

line 257: "... evolution from isotropic pores (IV15SH) to ..."

suggested:  "... evolution ... etc. ... pores (FREEZE-DRIED IV15SH) to ..."

Response: We have revised "...evolution from isotropic pores (IV₁₅SH) to "...evolution from isotropic pores (Freeze-dried IV₁₅SH) to ..."

10) (2.3. Characterization of the microstructures):

line 304/305: "... swollen state was also calculated."

suggested addition of short explanation:  "... swollen ... etc. ... calculated (USING SI-EQUATION 8, TAKING INTO ACCOUNT THE ABSORBED SWELLING WATER AS CONTRIBUTION TO THE AMORPHOUS PHASE)."

-related Fig. S.11 in SI file: a more descriptive scaling of the Y-axis would be from 0 to 8.0 (rather than 5.6 to 8.0).

Response: We have revised it to "... swollen state was also calculated." (using Equation 8 in the Supporting Information). Additionally, we have revised the scale of the y-axis in Fig. S.11 in SI file (from 0 to 8.0) (Figure R3-1).

Figure R3-1. Summary of average size of crystalline domains of IV₁₅SH, AV₁₅SH and AV₁₅SH-120.

11) line 323 (FTIR): it would be useful to specify, whether the FTIR spectra were recorded in dry or in the wet state.

Response: The FTIR spectra were recorded in the wet state, we have added relevant descriptions of the test conditions in the supporting information.

12) paragraph breaks would be suggested (as topic separation) at lines: 256, 285, 289, 297, and 302 (ideally, the first words in each paragraph should give a quick orientation).

Response: We have provided a quick orientation for the first words in some paragraph.

13) 2.4. Crack-propagation resistance:

line 344: Figure 4: possibly one representative graph (from the collection a–e) from Fig. S12 and one from Fig. S13 should be added to the composite Figure 4, because these cyclic characteristics are really interesting;

line 369: it would be interesting to add an eventual short comment comparing the different creep tendencies of the samples in Fig S13.

Response: Thank you very much for your suggestion. Figure 4. primarily discusses the notch-insensitivity and fatigue resistance of the hydrogel, as well as reveals the anti-fatigue mechanism of the hydrogel. Therefore, we believe that Figures S12 to S13 would be more appropriately placed in the supporting information. Figure S13 shows

cyclic stretching loading–unloading curves of different hydrogels at a predeterminate strain of 100 %, we did not obtain creep tendencies data.

14) The cyclic tests illustrated by Fig. S12 and Fig. S13 should be described in the Experimental Section of the Supporting Information File, as information about them is missing;

-were these tests conducted in H₂O ('underwater')?

Response: All tensile tests of the hydrogels with a dumbbell shape were conducted on a universal tensile machine (Instron Model 3367, USA) at room temperature with the speed of 50 mm/min. The thickness of the samples was measured using a digital measuring device and the width was measured using a vernier calipers. The area under the stress-strain curves to failure is defined as the “toughness”. The loading-unloading experiments under different applied strains (40%-240%) were conducted at room temperature with the speed of 50 mm/min. Cyclic loading-unloading test at the fixed strain of 100% was conducted underwater to study the dissipated energy of the hydrogels. We have described the tests in the Experimental Section of the Supporting Information.

To prevent dehydration of the hydrogel, we conducted the tests underwater. The relevant description has been added to the Supporting Information.

15) paragraph breaks would be suggested (as topic separation) at lines 358, 363, and 374 (ideally, the first words in each paragraph should give a quick orientation).

Response: We have revised the manuscript as suggested to ensure the first words in each paragraph were given a quick orientation.

16) 2.5. Anti-swelling performance and biocompatibility.

suggested: “Resistance to swelling” instead of “Anti-swelling performance” (or Anti-swelling properties/behavior, etc.).

Response: We have revised “Anti-swelling performance” to “Resistance to swelling” in the revised manuscript.

17) paragraph break would be suggested (as topic separation) at line 456, concerned with cell growth (ideally, the first words in the paragraph should give a quick orientation).

Response: We have revised the manuscript as suggested by clearly separating the content related to cell growth into distinct paragraphs and ensuring the opening sentence of each paragraph facilitates quick identification of the topic.

Dear Editors and Reviewers:

Thank you for your letter and for the reviewers' comments concerning our manuscript entitled "Hierarchical Crack-Resistant, Tissue-Mimetic Hydrogels Enabled by Progressive Nanocrystallization of Anisotropic Polymer Networks" (Manuscript ID: NCOMMS-25-30700B). We have incorporated the suggestions and comments in the revised manuscript and completed the checklist. These changes have been highlighted in red color. For easy reference, point-by-point responses to the comments of the reviewers are also given below in red color.

We hope we have been able to address the reviewer's concerns, and we would like to thank the reviewers and editors again for taking the time to provide their helpful input throughout the entirety of this review process.

Response to reviewers:

Reviewer #1 (Remarks to the Author):

The author has made revisions to the article based on the suggestions, and the article can be accepted.

Response: We sincerely appreciate your positive and valuable feedback. Thank you again for taking the time to provide helpful input throughout the entirety of this review process.

Reviewer #2 (Remarks to the Author):

In response to my earlier comments, the authors have made a number of useful revisions that strengthen the manuscript. They updated the process diagram to show the annealed state separately and explained more clearly that densification happens during salting-out, while the alignment is largely kept after soaking. They also added many results of intermediate samples to support this. In addition, they provided mechanical data normalized by polymer content, showing that the performance is still much better even when the effect of concentration is excluded (for example, the strength and toughness of AV20SH-120 are still far higher than IV10SH on a per-content basis). They corrected the orientation labels and added raw 1D SAXS curves in the SI for transparency. Overall, the revised paper is much stronger, and the authors now explain more clearly that combining organogel mechanical training, wet annealing, and citrate salting-out gives

anisotropic, crystallinity-reinforced PVA hydrogels with high fatigue threshold and a rare balance of strength, toughness, and swelling resistance. However, there are still the following detailed issues that require further improvement.

1. The manuscript contains too much supplementary information placed within parentheses, which interrupts the flow. Please consider streamlining by reducing the parenthetical content or integrating it into the main sentences.

Response: Thanks for this kind and valuable suggestion. We have streamlined the parenthetical content to improve readability.

2. In Figure S6, the modulus data of IV₁₀SH appear to be missing. If this is due to the values being too small to be clearly displayed, I recommend adding an enlarged inset bar chart of IV₁₀SH strength and modulus in the blank space of this figure to ensure data completeness and readability.

Response: Thanks for this kind and valuable suggestion. In Figure S6, the modulus data of IV₁₀SH could not be accurately presented in the existing graph due to its relatively small values. We have added an enlarged inset showing the modulus of IV₁₀SH in the blank area of this figure to ensure all critical information is clearly visible.

3. Please carefully check figure-level details for consistency and legibility. For example: Fig. 1(c) bars have outlines whereas other bar charts do not (style inconsistency); the central region of Fig. 2(i) is not readable; Fig. 4(f) bar chart shows black striping artifacts; error bar styles are inconsistent in Fig. S6 and Fig. S16(f); the numeric labels below Fig. S12 do not align with tick marks (please unify with the formatting used in Fig. S17(b)); in Fig. S23(a, d) the text color on the right side appears incorrect. Some of these may stem from source image quality; in any case, please re-export/replace assets and unify styles (e.g., bar fill without borders, consistent error-bar caps, tick/label alignment, adequate resolution or vector format).

Response: Thank you for your suggestion. We have revised each issue individually to enhance the quality of the source images.

Reviewer #3 (Remarks to the Author):

[Note from the Editor: Please see attached PDF]

The reviewer 3 is very impressed by the results presented by the authors (as stated previously).

The Authors also responded to all the points raised by the reviewer 3, and they also added interesting and valuable additional data and discussion, while responding to the reviewers 1 and 2.

After reading the revised versions of the Manuscript and SI file,

The reviewer 3 strongly suggests one correction (Figure 1a and Experimental: Preparation description) which is specified in an attached File in MS Word, which is important for the readers orientation in the family of the studied products and intermediates.

Response: Many thanks for the kind and valuable comments. We have revised Figure 1 and the preparation description in the Experimental Section according to your suggestions. But regarding the revisions to the preparation diagrams, we have partially adopted your suggestions.

For the authors consideration, the reviewer 3 also suggests two eventual short comments to be possibly added to the discussion (points 2 and 3).

Response: Many thanks for the kind and valuable comments. We have added two eventual short comments.

Technical: in some cases, the presented values and standard deviations should be rounded, e.g. " $268.67 \pm 48.22\%$ "  " $270 \pm 50\%$ ", or at least " $269 \pm 48\%$ ".

Response: Thank you for your suggestion. We have revised them in the manuscript.

1) Improvement of

Figure 1a is strongly suggested, in order to display all the discussed products and intermediates, as outlined graphically in the MS Word file attached by the reviewer 3. The related Experimental description in the Main Manuscript also should be improved as suggested in the mentioned File attached by the reviewer 3. The suggested corrections only modestly alter the actual image and the experimental description, while the reader will obtain a good orientation concerning the discussed products and intermediates, especially in view of the newly added discussion of some of the interesting intermediates.

Response: Many thanks for the kind and valuable comments. We have revised Figure 1 and the preparation description in the Experimental Section according to your suggestions. But regarding the revisions to the preparation diagrams, we have partially adopted your suggestions.

2) Discussion of importance, sequence and synergy of the gel modification steps:
(lines 93-101):

Here, in the revised Discussion, the authors very now very well describe the synergy and the sequence

of the applied modification steps.

For the authors' consideration:

> It also might be useful to point out that, the employed modification steps were applied to the gels in the sequence of their decreasing 'aggressivity'. In spite of the mildness of the (last) salting-out step, its effect on the mechanical properties (tensile tests), as well as on the crystallinity (XRD) was very pronounced. The kosmotropic and multiply hydrogen-bonding citrate anion appears to be very helpful in this final reorganization step where it causes additional crystallization and favors a defect-free structure (witnessed by XRD: the relatively strong 200 reflection at 22.3deg).

> The gradual ordering of the structure of the anisotropic hydrogels crosslinked by hydrogen bonded nanocrystallite domains could be commented in analogy to the comparison of irregular covalently crosslinked hydrogels (divinyl co-monomers, free radical polymerization) vs. hydrogels with an ideally homogeneous network structure (achieved by special synthesis routes). The regular network structure leads to much improved tensile properties, especially toughness. In the studied case, the physical crosslinks in the IVx gels are rather randomly distributed, similarly like the network junctions in the irregular covalently crosslinked hydrogels. Literature examples about the covalent hydrogels include [Sakai et al., *Macromolecules*, 2008, 41, 5379–5384. DOI: 10.1021/ma800476x] for the ideally homogeneous ones, or e.g. [Dusek et al., *Polymer Bulletin*, 1980, 3, 19–25. DOI: 10.1007/BF00263201], [Shibayama, *Macromol. Chem. Phys.* 1998, 199, 1–30. DOI: [https://doi.org/10.1002/\(SICI\)1521-3935\(19980101\)199:1<1::AID-MACP1>3.0.CO;2-M](https://doi.org/10.1002/(SICI)1521-3935(19980101)199:1<1::AID-MACP1>3.0.CO;2-M)] for the irregular ones.

Response: Many thanks for the kind and valuable comments. In the previous revised Manuscript, we have provided a detailed explanation of the role of each processing step and the changes in the crystalline structure.

In this study, the prepared IVxH hydrogel is an isotropic hydrogel formed through physical cross-linking. In terms of network structure orderliness, its physical cross-linking points (hydrogen-bonded nanocrystalline domains) exhibit a relatively random

distribution within the gel. For irregular covalently cross-linked hydrogels, the cross-linking points are randomly distributed in the network, which may lead to issues such as uneven pore sizes and numerous defects in the final network structure, thereby affecting their mechanical properties. In contrast, for an ideal homogeneous network structure hydrogel, the cross-linking points are uniformly distributed within the network, resulting in a regular and defect-minimized structure. Consequently, such hydrogels demonstrate superior tensile performance.

3) Tensile properties: Figure 2 a, d:

A comment might be useful (no additional graphs or experiments) about eventual elastic and plastic regions (deformation limits) of the tensile curves, in relation to high-slope low and slope regions in the tensile graphs.

In this context: How large is the visually observed approximate creep value after sample disruption (if the pieces are put together; the results in Fig. S 21 might suggest some creep)? E.g. is the lower-slope region at the high elongation values predominantly plastic (e.g. disconnection or reorganization of less perfect crystallites)? The relation of the individual preparation steps to the eventual elastic / plastic deformation regions might be interesting.

Response: Thanks for the insightful comments regarding the tensile behavior of the hydrogel and its relation to elastic/plastic deformation and possible creep. As a typical polymer-based material, the hydrogel exhibits viscoelastic characteristics. In a single tensile test (as shown in Fig. 2d), the stress - strain curve is predominantly governed by elastic deformation, and no distinct yield point is observed before fracture even in the lower-slope region at high elongations. It should be noted that creep, by definition, refers to time-dependent deformation under constant stress, which cannot be directly assessed through a standard tensile test with continuously increasing strain. However, after fracture, when the broken pieces are reassembled, a certain degree of permanent deformation is visually observed, indicating that irreversible changes have occurred. The extent of this permanent deformation depends on the specific structure and composition of the hydrogels.

In cyclic tensile tests (Fig. S21), hysteresis loops are clearly visible, and the strain is not fully recovered after each cycle, suggesting the presence of inelastic deformation.

This behavior can be attributed to molecular chain alignment, disruption and reorganization of hydrogen bonds, and other intermolecular interactions. Upon unloading, the chains cannot fully return to their initial configurations, indirectly supporting the notion of time-dependent viscoelastic flow or creep-like behavior under high strain.

The multi-step reinforcement process used in hydrogel preparation certainly influences the final mechanical performance, including the balance between elastic and plastic deformation. Each preparation step modifies the network structure, affecting how the material responds to deformation, particularly in terms of how reversible (elastic) or irreversible (plastic) the deformation becomes near the fracture point.

In our follow-up studies, we plan to systematically investigate the creep behavior—for example, by examining deformation under constant load, the effect of temperature, and how each preparation step influences the long-term viscoelastic performance.

Recommendation: Reject

The manuscript proposes a “stepwise construction” strategy combining mechanical training, wet annealing, and salting-out was proposed to fabricate PVA hydrogels with hierarchical, anisotropic, and biomimetic structures. The reported mechanical properties (e.g., strength > 60 MPa, toughness > 100 MJ/m³, fracture energy > 85 kJ m⁻² and fatigue threshold >2000 J/m²) are outstanding, representing a notable advance in tough hydrogel research. The authors attempt to elucidate structure–property relationships across multiple length scales (SEM, SAXS, WAXS, XRD and DSC) and present a schematic mechanism, reflecting a desire for in-depth understanding.

However, the manuscript faces several critical issues:

1. Lack of rigor in data presentation and interpretation: Including inconsistencies between text and figures, incorrect legends, and imprecise terminology.
2. Insufficient support for key claims: The SAXS/WAXS/XRD analyses lack depth, and some critical intermediate states are not directly characterized, leaving the stepwise construction mechanism incomplete.
3. Conceptual novelty: While the performance is impressive, the methods themselves (mechanical training, annealing, salting-out) are not new. The manuscript should Clarify their unique synergy or irreplaceability in achieving this superior performance.

In conclusion, the manuscript is not yet suitable for publication in *Nature Communications*.

There are some concerns (listed below) that should be addressed:

1. The authors describe a “stepwise construction” process, but the structural changes after mechanical training, then after annealing, and the structural evolution from salting-out should be clearly illustrated. In Figure 1a, “Mechanical training” and “Annealing” are combined into a single arrow. If annealing (i.e., conformational optimization and crystallinity enhancement) is a critical intermediate step, separately depicting the state “Annealing” (i.e., following “Mechanical training”) would make the schematic more persuasive. Additionally, the transition from “anisotropic organogel” to “anisotropic hydrogel” in Figure 1a does not effectively depict the structural densification expected from the salting-out process.
2. The schematic clearly labels a “soaked in water” step and depicts distinct microstructural differences between the post-salting-out state (pink) and the final hydrogel (green). However, this structural transition is not discussed in the text. This omission could lead readers to question the “structure locking” mechanism: specifically, at which step does structural densification actually occur? If densification happens during this “soaked in water” phase, rather than during “salting-out,” it would contradict the authors proposed core mechanism of “structure locking” by Hofmeister effect-driven salting-out.
3. It is noticed that the water content changes significantly after the “stepwise construction”. The mechanical properties of hydrogels are closely related to their water content, as a decrease in water content (i.e., an increase in polymer concentration) inherently leads to increased strength and modulus. To demonstrate that the observed enhancements arise from the proposed structural optimization rather than simply from concentration effects, it is recommend providing mechanical property data normalized by polymer content, along with a comparative analysis and discussion of the results before and after normalization.
4. In Figure 3c, the 2D SAXS and WAXS patterns of the AV₁₅SH-120 sample show clearly

different scattering orientations: the SAXS signal is stronger in the vertical direction, while the WAXS signal is more concentrated horizontally. This appears inconsistent with Figures 3d and 3f. Could the authors clarify whether this arises from differences in data processing or from orientation differences at different structural length scales? Additionally, it is suggest indicating the stretching direction in Figure 3c.

5. In Figure 3e, for AV₁₅SH-120, there's another smaller peak around $q = 1.625 \text{ \AA}^{-1}$. It is suggest discussing whether this peak may originate from another crystallographic plane or a higher-order reflection, as this would help readers better understand the evolution of the crystalline structure. Furthermore, the q -values of IV₁₅SH and AV₁₅H appear very close in the figure. Could the authors clarify the method used (e.g., peak fitting) to accurately distinguish the q_{max} of these two peaks? Providing the fitting results or explain more about how you found these peak positions would strengthen the reliability of the conclusions.
6. Figure 3g presents the Lorentz-corrected (Iq^2) 1D SAXS profiles, which is effective for analyzing long-period structures (L) and the evolution of lamellar features. However, However, to offer readers a more comprehensive understanding of the sample's original scattering information and potentially reveal other structural features, it is recommended authors providing the raw, uncorrected 1D SAXS scattering curves, either in the main text or the Supporting Information. Such an addition typically helps readers better grasp the overall characteristics of the scatterers and the original state of the data before processing.
7. It is recommended that the authors clearly indicate whether the 1D SAXS/WAXS profiles are obtained from integration parallel or perpendicular to the nanofiber alignment direction, as the anisotropic structure would yield significantly different scattering features in each orientation. It is further recommended that the authors include specific details on the integration method and direction used for this data in all relevant 1D SAXS/WAXS figure captions and corresponding descriptions in the main text.
8. The authors interpret “ L ” as the average distance between adjacent crystalline domains, stating that the decrease in L , defined as the decrease in average distance between adjacent crystalline domains, as evidence of polymer chain densification. While this interpretation is plausible, L can also be influenced by changes in crystal morphology, grain size, or the number of crystalline nuclei. Considering this complexity, attributing the decrease in L directly and solely to “densification between polymer chains” may lack sufficient rigor. It is suggested the authors consider incorporating WAXS data to provide more specific and direct evidence for this claim of “densification between polymer chains”.
9. In the XRD patterns shown in Figure 3i, AV₁₅SH-120 exhibits not only a sharp and intense main peak at $2\theta \approx 19.7^\circ$, but also a secondary peak at higher diffraction angles (e.g., around $2\theta \approx 22\text{-}23^\circ$), similar to what is observed in the WAXS data. This observation prompts the question of whether new, or significantly enhanced, crystalline plane has formed in the PVA gel after the series of treatments involving mechanical training, high-temperature annealing, and salting-out. It is suggested that the authors provide a more detailed discussion and assignment of this secondary peak in the manuscript. Furthermore, the possible relationship between the emergence of this new crystalline plane and the observed exceptional mechanical properties warrants further exploration.

10. The manuscript refers to the formation of “proto-crystalline regions” and state that the final material possesses “crystalline domains”. It is would appreciate clarification on how do the authors define “proto-crystalline regions”? What are the specific structural distinctions between these and the more stable “crystalline domains” ultimately observed? Furthermore, can the existing characterization data (e.g., XRD/WAXS) differentiate or provide evidence for the existence and evolution of these distinct crystalline states? It is observed that neither the schematic for the fabrication process (Figure 1a) nor that for structural characterization (Figure 3) clearly depicts “proto-crystalline regions” or their transformation into the final “crystalline domains”. Therefore, it is suggest considering revisions to these schematics (Figure 1a) to more explicitly illustrate the evolution of the crystalline structure.
11. The manuscript emphasizes a “stepwise construction” strategy involving mechanical training, annealing, and salting-out. While the authors compare the structures of different final states (e.g., IV₁₅SH, AV₁₅H, AV₁₅SH-120), it is strongly recommended that they also include structural characterization of intermediate structures after each key processing step: (1) post-mechanical training only; (2) post-training and annealing; and (3) post-training, annealing, and salting-out. To achieve this, it would be beneficial to provide characterization data (such as SEM, WAXS, SAXS, XRD, and DSC). This approach would more clearly establish the direct link between each processing stage and the resulting structural changes, thus offering more robust support for “stepwise construction” mechanism.
12. While the fatigue threshold (G_0) is a standard concept, the authors do not explicitly state in the manuscript how they specifically defined or extracted this value from the dc/dN vs. G data in Figures 4a-e. For instance, it is unclear whether G_0 was determined by extrapolation to $dc/dN=0$. Furthermore, the number of data points in the plots appears limited, the linear fits are not convincing, and some exhibit poor linearity, and the overall fitting accuracy seems questionable.
13. For all relevant mechanical property test results (e.g., tensile strength, fracture energy, fatigue tests, etc.), it is also recommended that the authors specify whether the tests were conducted parallel or perpendicular to the nanofiber orientation.

The manuscript still contains numerous instances of imprecise or nonstandard expressions:

1. Line 119, in the sentence “Hofmeister-specific ion coordination during salting-out,” the use of the term “coordination” is inappropriate. The Hofmeister effect primarily refers to how specific salt ions influence polymer solubility, aggregation behavior, and phase states by modulating solvent structure (especially water) and non-covalent interactions between solutes (such as electrostatic, hydrophobic, and hydrogen bonding interactions). This process typically does not involve the formation of coordination bonds in a chemical sense.
2. Line 174, The statement “Wet annealing introduces temperature-dependent conformational optimization” is not sufficiently rigorous. Based on the overall context of the manuscript, wet annealing not only facilitates conformational rearrangements but also leads to structural changes.
3. In Figure 2a, the sample name is incorrect: “AV₁₅H-120” should be “AV₁₅SH-120”.

4. In Figure 3e, the authors state that, “IV₁₅SH、 AV₁₅H and AV₁₅SH-120 appear at 0.46 nm, 0.43 nm, and 0.42 nm, respectively”. However, in the figure, the authors have labeled the strongest peak indicated was 0.46 nm for AV₁₅SH-120.

Suggested corrections in Figure 1a:

Suggested corrections in the Experimental Section:

highlighting: ~~deleted text~~ added text

603 **(1) Preparation of isotropic hydrogels:** PVA powder was dissolved in water and
 604 stirred at 100 °C for 5 h to obtain a homogeneous 15 wt% solution. The solution was
 605 then poured into polytetrafluoroethylene (PTFE) molds, cooled to room temperature,
 606 and subsequently frozen at -18 °C for 2 h. The resulting hydrogels were soaked in a
 607 glycerol/ethanol solvent mixture for 48 h, thus obtaining the isotropic organogel IV_x (where x
 indicates the initial PVA mass fraction).

Some of the produced IV_x specimens were processed to obtain a modified form of this intermediate:

~~The obtained organogels were then~~ They were immersed

608 in a 1.5 M sodium citrate solution for 12 h, followed by re-immersion in deionized
 609 water to yield PVA hydrogels, denoted as IV_xSH , where x indicates the initial PVA mass
 610 fraction.

611 **(2) Preparation of trained anisotropic gels:** The IV_x organogels prepared as described in (1)
 612 underwent mechanical training for 30 cycles at a strain of 300% and a tensile rate of
 613 200 mm/min, under ambient conditions, thus obtaining the 'trained' anisotropic organogel AV_x
 (where x indicates the initial PVA mass fraction). Some of the produced AV_x specimens were
 processed to obtain a modified form of this intermediate: ~~The trained gel (AV_x) was then directly~~
 They were soaked in deionized water for 48

614 h to obtain thus yielding the hydrogels AV_xH , where x indicates the initial PVA mass fraction.

615 **(3) Preparation of trained and annealed anisotropic gels:** The AV_x organogels prepared
 616 as described in (2) underwent mechanical training for 30 cycles at a strain of 300% and a
 617 tensile rate of 200 mm/min. Then, wet annealing was performed were subjected to wet
 annealing at 120 °C for 30 min

618 to obtain $AV_x-y^\circ C$, where x indicates the initial PVA mass fraction, and y denotes the wet

619 annealing temperature. ~~The~~ Some of the produced gels ($AV_{x-y}^{\circ C}$) were processed to obtain a
620 modified form of this intermediate: ~~was then directly~~ They were soaked in deionized water for
621 48 h to obtain $AV_xH-y^{\circ C}$.

621 **(4) Preparation of trained, annealed and salted out anisotropic gels:** The ~~annealed and trained~~
622 anisotropic organogels $AV_{x-y}^{\circ C}$ prepared as described in (3) ~~underwent mechanical training for 30 cycles~~
623 ~~at a~~ strain of 300% and a tensile rate of 200 mm/min. Then, wet annealing was performed
624 ~~at 120 °C for 30 min. Subsequently, it was~~ were immersed in a sodium citrate solution for 12
625 h to obtain $AV_xS-y^{\circ C}$, followed by soaking in water for another 48 h ~~to obtain~~ which yielded the
626 final product $AV_xSH-y^{\circ C}$,

626 where x denotes the initial PVA mass fraction, and y denotes the wet-annealing
627 temperature.

628

629 The remaining test and characterization methods can be found in the supporting
630 information.

631

632 **Acknowledgements**